# CRYPTOCHROMES confer robustness, not rhythmicity, to circadian timekeeping

Marrit Putker[1] (iD), David C S Wong[1] (iD), Estere Seinkmane[1] (iD), Nina M Rzechorzek[1] (iD), Aiwei Zeng[1] (iD), Nathaniel P Hoyle[1], Johanna E Chesham[1] (iD), Mathew D Edwards[1] (iD), Kevin A Feeney[1] (iD), Robin Fischer[2], Nicolai Peschel[2] (iD), Ko-Fan Chen[3] (iD), Michael Vanden Oever[4] (iD), Rachel S Edgar[4] (iD), Christopher P Selby[5], Aziz Sancar[5] (iD) & John S O'Neill[1,*] (iD)

## Abstract

Circadian rhythms are a pervasive property of mammalian cells, tissues and behaviour, ensuring physiological adaptation to solar time. Models of cellular timekeeping revolve around transcriptional feedback repression, whereby CLOCK and BMAL1 activate the expression of PERIOD (PER) and CRYPTOCHROME (CRY), which in turn repress CLOCK/BMAL1 activity. CRY proteins are therefore considered essential components of the cellular clock mechanism, supported by behavioural arrhythmicity of CRY-deficient (CKO) mice under constant conditions. Challenging this interpretation, we find locomotor rhythms in adult CKO mice under specific environmental conditions and circadian rhythms in cellular PER2 levels when CRY is absent. CRY-less oscillations are variable in their expression and have shorter periods than wild-type controls. Importantly, we find classic circadian hallmarks such as temperature compensation and period determination by CK1δ/ϵ activity to be maintained. In the absence of CRY-mediated feedback repression and rhythmic *Per2* transcription, PER2 protein rhythms are sustained for several cycles, accompanied by circadian variation in protein stability. We suggest that, whereas circadian transcriptional feedback imparts robustness and functionality onto biological clocks, the core timekeeping mechanism is post-translational.

**Keywords** cellular clock; circadian rhythm; cryptochrome; daily timekeeping; robustness
**Subject Category** Signal Transduction
**The EMBO Journal (2021) 40: e106745**

## Introduction

The adaptive advantage conferred on organisms by anticipation of the 24-h cycle of day and night has selected for the evolution of circadian clocks that, albeit in different molecular forms, are present throughout all kingdoms of life (Rosbash, 2009; Edgar *et al*, 2012). Circadian rhythms are robust, in that they are "capable of performing without failure under a wide range of conditions" (Merriam-Webster Dictionary, 2020). The mechanism proposed to generate daily timekeeping in mammalian cells is a delayed transcriptional–translational feedback loop (TTFL) that consists of activating transcription factor complexes containing CLOCK and BMAL1 and repressive complexes, containing the BMAL1:CLOCK targets PERIOD and CRYPTOCHROME (reviewed in Dunlap, 1999; Reppert & Weaver, 2002; Takahashi, 2016). Various coupled, but non-essential, auxiliary transcriptional feedback mechanisms are thought to fine-tune the core TTFL and co-ordinate cell-type-specific temporal organisation of gene expression programs; the best characterised being effected by the E-box mediated rhythmic expression of REV-ERBα/β, encoded by the Nr1d1/2 genes (Preitner *et al*, 2002; Ueda, 2007; Liu *et al*, 2008; Takahashi, 2016). These auxiliary loops are not considered sufficient to generate circadian rhythms in the absence of the core TTFL (Preitner *et al*, 2002; Liu *et al*, 2008).

CRY1 and CRY2 operate semi-redundantly as the essential repressors of CLOCK/BMAL1 activity (Ye *et al*, 2014; Chiou *et al*, 2016), required for the nuclear import of PER proteins, and together are considered indispensable for circadian regulation of gene expression *in vivo* as well as in cells and tissues cultured *ex vivo* (Kume *et al*, 1999; Sato *et al*, 2006; Chiou *et al*, 2016; Ode *et al*, 2017). Certainly, mice homozygous null for *Cry1* and *Cry2* do not express circadian behavioural rest/activity cycles under standard experimental conditions (Thresher *et al*, 1998; Horst & Muijtjens, 1999; Vitaterna *et al*, 1999).

1   MRC Laboratory of Molecular Biology, Cambridge, UK
2   Biozentrum Universität, Würzburg, Germany
3   Institute of Neurology, University College London, London, UK
4   Faculty of Medicine, Imperial College London, London, UK
5   Department of Biochemistry and Biophysics, University of North Carolina School of Medicine, Chapel Hill, NC, USA
    *Corresponding author. Tel: +44 7739 729425; E-mail: oneillj@mrc-lmb.cam.ac.uk
    †Present address: UCL Sainsbury Wellcome Centre for Neural Circuits and Behaviour, London, UK
    ‡Present address: Department of Genetics and Genome Biology, University of Leicester, Leicester, UK

The hypothalamic suprachiasmatic nucleus (SCN) is a central locus for circadian co-ordination of behaviour and physiology, and research over the last two decades has stressed the strong correlation between SCN timekeeping *in vivo* and its activity when cultured *ex vivo* (Welsh *et al*, 2010; Anand *et al*, 2013). We were therefore intrigued by the observation that roughly half of organotypic SCN slices prepared from homozygous $Cry1^{-/-},Cry2^{-/-}$ (CRY knockout; CKO) mouse neonates continue to exhibit ~ 20 h (short period) rhythms, observed using the genetically encoded PER2::LUC clock protein::luciferase fusion reporter (Maywood *et al*, 2011; Ono *et al*, 2013b), despite having previously been described as arrhythmic (Liu *et al*, 2007). Moreover, short period circadian rhythms of locomotor activity have previously been reported for CKO mice raised from birth under constant light (Ono *et al*, 2013a). As CKO SCN oscillations were only observed in cultured neonatal organotypic slices *ex vivo*, they were suggested to be a network-level SCN-specific rescue by the activity of neuronal circuits, that desynchronise during post-natal development (Welsh *et al*, 2010; Ono *et al*, 2013b). In our view, however, these observations are difficult to reconcile with an essential requirement for CRY in the generation of circadian rhythms. Rather, they are more consistent with CRY making an important contribution to circadian rhythm stability and functional outputs, rather than to the timekeeping mechanism *per se*, as recently shown for the genes *Bmal1* and *Clock* (Landgraf *et al*, 2016; Ray *et al*, 2020), which had both previously been thought indispensable for circadian timekeeping in individual cells (Bunger *et al*, 2000; DeBruyne *et al*, 2007). This is further supported by reports that constitutive expression of *Cry1* in cells and SCN perturbs but does not abolish circadian oscillations (Fan *et al*, 2007; Chen *et al*, 2009; Nangle *et al*, 2014; Edwards *et al*, 2016).

Recent observations have further questioned the need for transcriptional feedback repression to enable cellular circadian timekeeping. For example, circadian protein translation is regulated by cytosolic BMAL1 through a transcription-independent mechanism (Lipton *et al*, 2015), and isolated erythrocytes exhibit circadian rhythms despite lacking any DNA (O'Neill & Reddy, 2011; Cho *et al*, 2014). Moreover, circadian timekeeping in some species of eukaryotic alga and prokaryotic cyanobacteria can occur entirely post-translationally (Sweeney & Haxo, 1961; Nakajima *et al*, 2005; Tomita *et al*, 2005; O'Neill *et al*, 2011). Whether non-transcriptional clock mechanisms operate in other (nucleated) mammalian cells is unknown however, and hence their mechanism and relationship with TTFL-mediated rhythms is an open question.

Here, we used cells and tissues from CRY-deficient mice, widely accepted not to exhibit circadian transcriptional regulation (Kume *et al*, 1999; Ukai-Tadenuma *et al*, 2011; Edwards *et al*, 2016) to test whether any timekeeping function remained from which we might begin to dissect the mechanism of the postulated transcription-independent cytosolic oscillator or "cytoscillator" (Hastings *et al*, 2008).

## Results

### Cell-autonomous circadian PER2::LUC rhythms in the absence of CRY proteins

Consistent with previous observations, we found no significant circadian organisation of locomotor activity in CRY-deficient (CKO) mice following entrainment to 12 h:12 h light:dark (LD) cycles or in constant light (LL). Upon transition from constant light to constant darkness (DD) [described to be a stronger zeitgeber (Chen *et al*, 2008)] however, CKO mice expressed rhythmic bouts of consolidated locomotor activity with an average period of ~ 17 h and greater variance than WT controls (Figs 1A and B, and EV1A–C). In Fig 1, representative actograms are plotted as a function of endogenous tau ($\tau$) to allow the periodic organisation of rest–activity cycles to be readily observed; 24-h-plotted actograms are shown in Figs EV1A and EV2A. CKO rhythms under these conditions showed significantly reduced period and amplitude compared with wild-type (WT) controls, but persisted for > 2 weeks, consistent with these mice possessing a residual endogenous biological oscillation that is not entrained by standard environmental light:dark cycles (Fig EV2A–C and Appendix). In support of this interpretation, and in accordance with previous reports (Maywood *et al*, 2011; Ono *et al*, 2013b), longitudinal bioluminescence recordings of organotypic PER2::LUC SCN slices cultured *ex vivo* from WT or CKO neonates revealed rhythmic PER2 expression in approximately 40% of CKO slices (Fig 1C). In line with behavioural data and previous reports, these CKO SCN rhythms exhibited significantly shorter periods compared with WT controls (Fig 1D).

Two explanations might account for the variable CKO SCN phenotype: (i) the previously proposed explanation: genetic loss of function is compensated at a network level by SCN-specific neuronal circuits whose function is sensitive to developmental phase and small variations in slice preparation (Liu *et al*, 2007; Evans *et al*, 2012; Ono *et al*, 2013b; Tokuda *et al*, 2015); or (ii) CKO (SCN) cells have cell-intrinsic circadian rhythms that are expressed (or observed) more stochastically and with less robustness than their WT counterparts, and can be amplified by SCN interneuronal signalling (Welsh *et al*, 2010; O'Neill & Reddy, 2012).

To distinguish between these two possibilities, we asked whether PER2::LUC rhythms are observed in populations of immortalised PER2::LUC CKO adult fibroblasts, which lack the specialised interneuronal neuropeptidergic signalling that is so essential to SCN amplitude and robustness *in* and *ex vivo* (Welsh *et al*, 2010; O'Neill & Reddy, 2012). We observed this to be the case (Figs 1E, and EV1C and D). Across > 100 recordings, using independently generated cell lines cultured from multiple CRY-deficient mice (male and female), we observed PER2::LUC rhythms that persisted for several days under constant conditions. Again, the mean period of rhythms in CRY-deficient cells was significantly shorter than WT controls, and with increased variance within and between experiments (*F*-test *P*-value < 0.0001, Figs 1F, and EV1E and F). Consistent with SCN results, rhythmic PER2::LUC expression in CKO cells occurred stochastically between experiments, being observed in ~ 30% of independently performed assays. Importantly, there was very little variation in the occurrence of rhythmicity within experiments meaning that in any given recording all CKO replicate cultures were rhythmic or none, whereas WT cultures were always rhythmic. CKO PER2::LUC rhythms damped more rapidly than wild-type controls (Fig EV1G), and were more sensitive to acute changes in temperature than WT controls (Fig 2A and C), consistent with their oscillation being less robust. Crucially though, the PER2::LUC rhythms in CKO cells were temperature-compensated (Fig 2A and B) and entrained to 12 h:12 h 32°C:37°C temperature cycles in the same phase as WT controls (Fig 2C), and thus conform to the classic

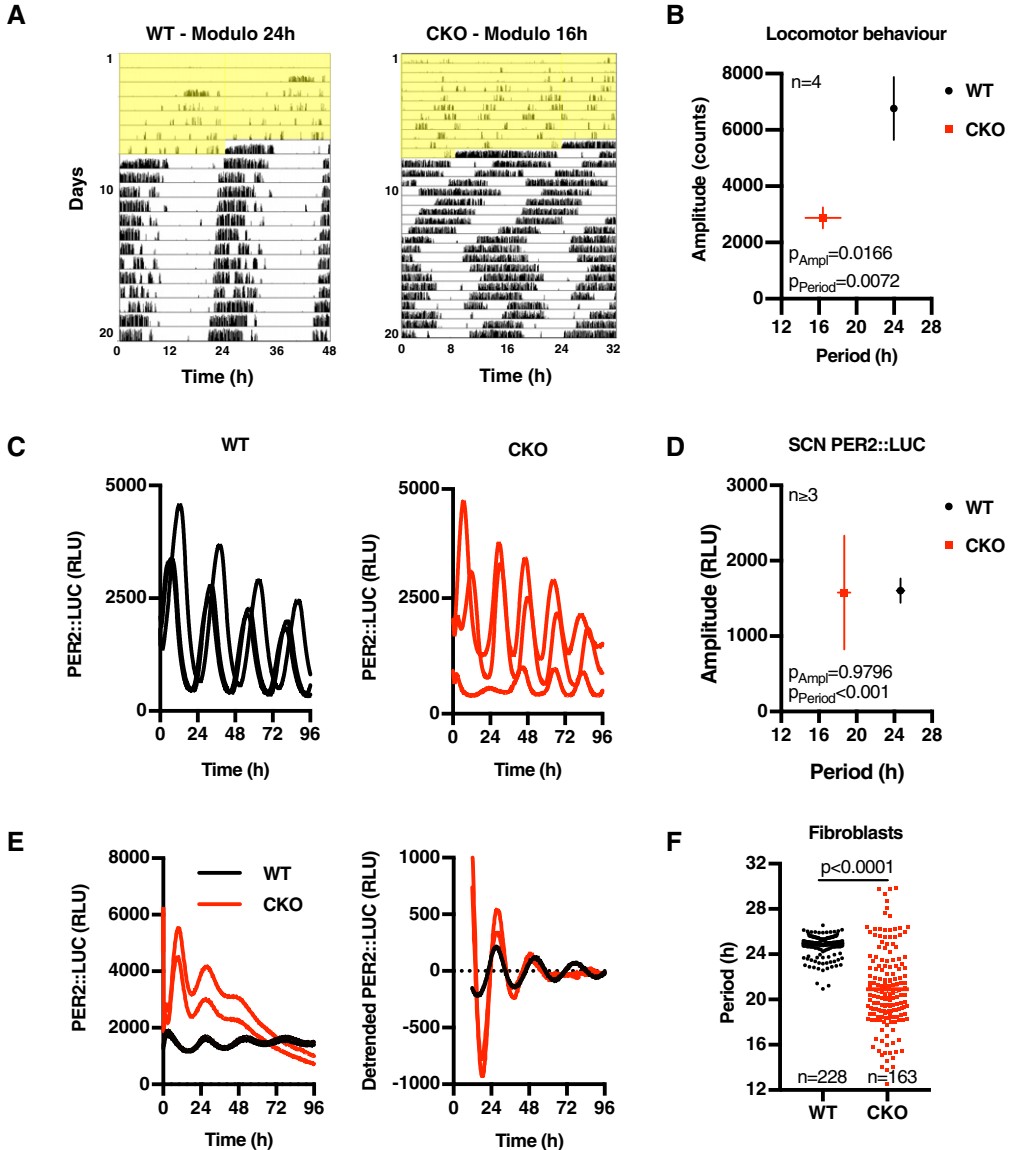

**Figure 1.  CRY-independent circadian timekeeping occurs cell-autonomously.**

A   Representative double-plotted actograms showing wheel-running activity of wild-type (WT) and CRY-deficient (CRY knockout; CKO) mice during constant light (yellow shading) and thereafter in constant darkness. Note the 48 h x-axis for WT vs. 32 h for CKO. Full figure showing CKO data in modulo 24 h is presented in Fig EV1A.

B   Mean period and amplitude (± SEM) of mouse behavioural data (*n* = 4). *P*-values were calculated by two-way ANOVA.

C   Longitudinal bioluminescence recordings of organotypic SCN slices from WT (black) and CKO (red) PER2::LUC mice (RLU; relative light units).

D   Mean period and amplitude (± SEM) of rhythmic SCN bioluminescence traces. *P*-values were calculated by two-way ANOVA.

E   Circadian PER2::LUC expression in immortalised WT and CKO adult lung fibroblasts. Left panel shows two raw traces of a representative longitudinal bioluminescence recording, and right panel shows same data detrended with a 24-h moving average to remove differences in baseline expression.

F   Period of rhythmic fibroblast bioluminescence traces from at least 31 experiments (*n* ≥ 3 per experiment, individual values ± SEM shown). *P*-values were calculated by an unpaired *t*-test with Welch correction. Standard deviations differ significantly between WT and CKO (*F*-test: *P* < 0.0001).

definition of a circadian rhythm (Pittendrigh, 1960), which does not stipulate any lower threshold for amplitude or robustness.

Clearly then, the *bona fide* circadian rhythms we observed in cultured CKO cells are insufficiently robust to facilitate entrainment of ~ 24-h rest–activity cycles that are a classic hallmark of circadian rhythms *in vivo*. This suggests that CRY-dependent transcriptional feedback repression confers robustness to rhythmic cellular clock

output that is required for circadian organisation of overt behaviour, rather than generating circadian rhythms *per se*. To test this in another model system, we turned to *Drosophila melanogaster*, where Timeless fulfils the functionally analogous role to mammalian CRY proteins as the obligate partner of Period, required for repression of circadian transcription at E-box promotor elements, and is required for the circadian organisation of locomotor activity

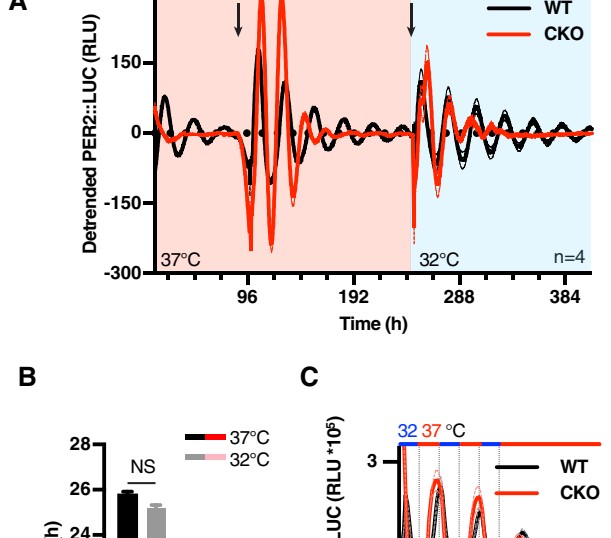

**Figure 2. CRY-less oscillations are temperature-compensated and entrained.**

A  Detrended traces of bioluminescence recordings of WT and CKO fibroblast at different constant temperature conditions within the physiological range ($n$ = 4, solid lines: mean, dashed lines: ± SEM). Temperature was changed from 37°C to 32°C halfway through the experiment, as depicted by red/blue shading. Arrows represent medium changes. Note the lack of rhythmicity in the first 3 days in CKO and the appearance of rhythmicity after the first medium change.

B  Quantification of period from recordings presented in (A). Both WT and CKO oscillations are temperature-compensated with respective $Q_{10}$s of 1.05 and 0.95 ($n$ = 3, mean ± SEM). $P$-values were determined by two-tailed $t$-test.

C  Bioluminescence of WT and CKO PER2::LUC cells during temperature entrainment (12 h 32°C (blue)–12 h 37°C (red)) ($n$ = 3, solid lines: mean, dashed lines: ± SEM).

under constant darkness (Sehgal *et al*, 1994, 1995). In assays of Period::LUC (XLG-LUC) activity in freely behaving flies, we observed circadian bioluminescence rhythms in *timeless* knockout animals that were significantly longer in period than WT controls (Fig EV2D–F). As observed for CRY-deficient cells, rhythms in Timeless-deficient flies persisted over several days with much lower amplitude than WT flies.

In contemporary models of the mammalian cellular clockwork, CRY proteins are essential for rhythmic PER protein production; however, the stability and activity of PER proteins are also regulated post-translationally (Iitaka *et al*, 2005; Lee *et al*, 2009; Philpott *et al*, 2020). Considering recent reports that there is no obligate relationship between clock protein turnover and circadian regulation of its activity in the fungus *Neurospora crassa* (Larrondo *et al*, 2015), that nascent transcription is not required for circadian rhythms in the green lineage (O'Neill *et al*, 2011), or in isolated human red blood cells (O'Neill & Reddy, 2011), we next investigated the relative

contribution of transcriptional vs. post-translational regulation to circadian PER2::LUC rhythms in CRY-deficient cells.

## CRY-independent PER2::LUC rhythms are driven by a non-transcriptional process

CRY has previously been described as the driving factor for feedback repression of BMAL1/CLOCK-dependent transcriptional activation, and is therefore considered essential to the rhythmic regulation of clock-controlled genes (CCGs). In fact, overexpression studies have suggested PER requires CRY to exert its function as a BMAL1-CLOCK repressor (Ye *et al*, 2014; Chiou *et al*, 2016). This importance of CRY for BMAL1-CLOCK repression (and auto-repression of *Cry* and *Per*) was also suggested by the increased PER2::LUC levels observed in CKO cells (Figs 1E and EV1C). Indeed, at the peak of PER2::LUC expression, CKO cells contain approximately twice as many PER2 molecules compared with their WT counterparts (Figs 3 A and EV3A).

Although not sufficient to completely rescue rhythms in CKO cells, it seemed plausible that increased PER or other clock protein expression might partially compensate for the loss of CRY function and continue to exert auto-regulation through rhythmic BMAL1-CLOCK binding, thereby accounting for the residual PER2::LUC rhythms in CKO cells. To test this possibility, we compared BMAL1-PER2 binding at the expected peak of BMAL1-PER2 complex formation (i.e. at the peak of PER2::LUC expression) in WT and CKO cells. To this end, we immunoprecipitated BMAL1 and measured the associated PER2::LUC activity. In accordance with CRY being required for PER2-BMAL1 binding, we did not find a PER2::LUC-BMAL1 complex in CKO cells, whilst the complex was readily detected in WT cells (Figs 3B, and EV3B and I), strongly suggesting that residual oscillations in PER2::LUC cannot result from a residual negative feedback upon the BMAL1-CLOCK complex.

In the absence of PER:CRY-mediated feedback repression, it seemed unlikely that CRY-independent oscillations in PER2::LUC expression are driven directly by rhythms in *Per2* transcription. Indeed, whereas PER2::LUC in co-recorded cells showed a clear variation over 24 h, *Per2* mRNA in parallel replicate CKO cultures instead exhibited a gradual accumulation (Fig 3C). In contrast and as expected (Feeney *et al*, 2016a), *Per2* mRNA in WT cells varied in phase with co-recorded PER2::LUC oscillations. The gradual increase of *Per2* mRNA in CKO cells is concordant with *Per2* transcriptional derepression predicted by the canonical TTFL model, accounting for the generally increased levels of PER2::LUC we observed (Fig 3A), but not their oscillation. In agreement with these findings and in contrast with WT cells, *Bmal1* mRNA also showed no significant variation in CKO cells (Fig EV3C), suggesting that E-box-dependent circadian regulation of REV-ERB activity may not occur in the absence of CRY-mediated feedback repression. In an independent validation, we assessed the activity of the circadian E-box-driven Cry1-promoter (Maywood *et al*, 2013) in mouse adult WT and CKO lung fibroblasts (MAFs) (Fig EV3D), as well as the *Per2*- and *Rev-erbα*- (*Nr1d1*-) promoters in mouse embryonic fibroblasts (MEFs) (Figs 3D and EV3E–H). No rhythmic *Cry1*- or *Per2*-promoter activity was observed in either set of CKO cells under any condition, whereas isogenic control cells showed clear circadian regulation of these promoters.

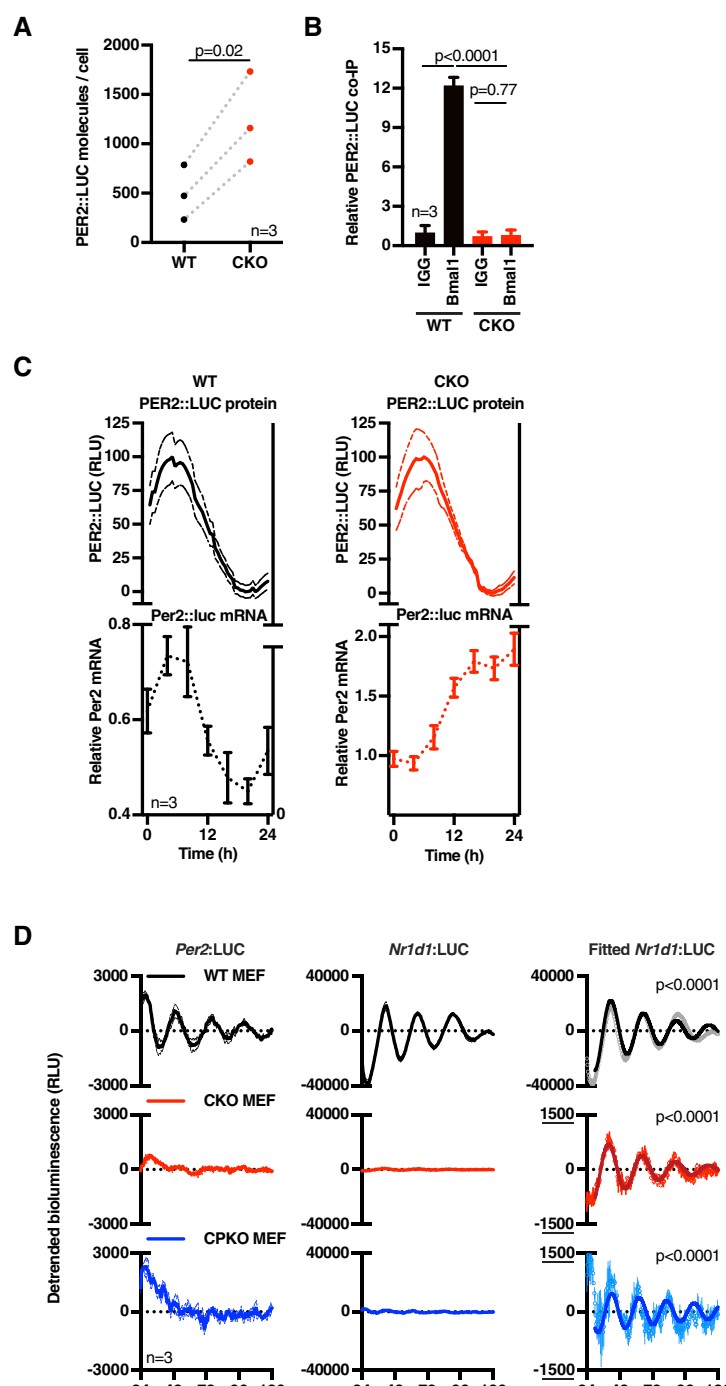

**Figure 3.  CRY-independent rhythms are regulated post-transcriptionally.**

A  Mean number of PER2::LUC molecules per cell at the estimated peak of PER2 expression for each cell line (mean of three experiments, *n* = 3 each). *P*-values were calculated by paired *t*-test.

B  PER2::LUC binding to BMAL1 in WT and CKO cells. Cells were harvested at the peak of PER2 expression, BMAL1 was immunoprecipitated, and PER2::LUC binding was measured by bioluminescence measurements (*n* = 3, mean ± SD). *P*-values were calculated by unpaired *t*-test.

C  Per2 mRNA levels in WT (left) and CKO (right) cells were determined by qPCR over one circadian cycle (bottom), whilst PER2::LUC bioluminescence (top, min-max normalised) was recorded from parallel cultures. Per2 mRNA reported relative to Rns18s (bottom), *n* = 3, ± SEM; PER2::LUC (top) presented as mean (solid) ± SEM (dashed), *n* = 3. The WT mRNA trace was preferentially fit by a circadian damped sine wave compared with straight line (*P* = 0.0412, extra sum-of-square F-test), whereas CKO data were not (ns).

D  Detrended *Per2* and *Nr1d1* promoter activity in WT, CKO and quadruple *Cry1/2-Per1/2* knockout (CPKO) mouse embryonic fibroblasts (MEFs) recorded at 37°C, *n* = 3, mean (solid) ±SEM (dashed). *Nr1d1* data were preferentially fit with a circadian damped sine wave over straight line (*P* < 0.0001, extra sum-of-squares *F*-test) (right hand graphs, solid lines; error bars, SEM). Similar recordings performed at 32°C and an expanded view of *Per2* data are presented in Fig EV3E and F.

In recordings from *Nr1d1*:LUC MEFs however, we were most surprised to observe temperature-compensated circadian rhythms in the activity of the Nr1d1 promoter in CKO cells, at just ~ 3% amplitude of WT cells, that persisted for several days (Figs 3D and EV3E, red traces). In the same experiments, similar but still noisier and lower amplitude rhythms were also detected in quadruple knockout MEFs that were also deficient for PER1/2, as well as CRY1/2 (CPKO, Figs 3D and EV3E, blue traces), confirming these oscillations cannot be attributable to any vestigial activity of PER proteins. We acknowledge it is conceivable that some unknown TTFL-type mechanism might generate these residual oscillations in *Nr1d1* promoter activity. However, we find it more plausible that residual oscillations of *Nr1d1*:LUC in CKO cells are the output of a post-translational timekeeping mechanism, from which the amplification and robustness conferred by CRY-dependent transcriptional feedback repression has been subtracted. Indeed, we note that besides CRY, *Nr1d1* expression is regulated by many other transcription factors, e.g. AP-1, NRF2, NF-KB and BMAL1/CLOCK (Preitner *et al*, 2002; Yang *et al*, 2014; Wible *et al*, 2018), whose activity is regulated post-translationally by the same rather promiscuous kinases that rhythmically regulate PER and BMAL1 in other contexts (Eide *et al*, 2002; Iitaka *et al*, 2005; Sahar *et al*, 2010; Narasimamurthy *et al*, 2018), e.g. casein kinase 1 and glycogen synthase kinase (Preitner *et al*, 2002; Liang & Chuang, 2006; Tullai *et al*, 2011; Rada *et al*, 2011; Medunjanin *et al*, 2016; Jiang *et al*, 2018).

### Circadian control of PER2 stability persists in the absence of CRY

The concentrations of luciferase substrates (Mg.ATP, luciferin, $O_2$) under our assay conditions are > 10× higher than their respective $K_m$ (Feeney *et al*, 2016a) and so it is implausible that PER2::LUC rhythms in CKO cells result from anything other than circadian regulation in the abundance of the PER2::LUC fusion protein. Indeed, PER2::LUC levels measured in cell lysates perfectly mirrored longitudinal PER2::LUC recordings from both WT and CKO cells (Fig 4A). We observed that the addition of the proteasomal inhibitor MG132 to asynchronous cells led to acute increases in PER2::LUC levels which were significantly greater in CKO cells than in WT controls, indicating that CKO cells support higher basal rates of PER2 turnover (Fig 4B and C). In consequence therefore, relatively small changes in the rate of PER2::LUC translation or degradation should be sufficient to affect the steady state PER2::LUC concentration. CKO cells exhibit no rhythm in *Per2* mRNA (Fig 3C and D), nor do they show a rhythm in global translational rate (Fig EV4A and B), nor did we observe any interaction between BMAL1 and S6K/eIF4 as occurs in WT cells (Lipton *et al*, 2015; Fig EV4C). We therefore investigated whether changes in PER2::LUC stability might be responsible for the persistent bioluminescence rhythms in CKO cells, by analysing the decay kinetics of luciferase activity during saturating translational inhibition.

In the presence of 10 μM cycloheximide (CHX), PER2::LUC bioluminescence decayed exponentially (Figs 4D and EV4D, $R^2 > 0.9$), with a half-life that was consistently < 2 h (Figs 4D and EV4D, F and G); much less than the half-life of luciferase expressed in fibroblasts under a constitutive promoter (≥ 5 h, Fig EV4D, E and H). Moreover, we observed a significant variation (±50%) in the half-life of PER2::LUC between the rising and falling phases of its expression (1.5 vs. 1 h, respectively, Figs 4D and EV4G) without

any commensurate change in global protein turnover (Fig EV4H). Strikingly, we also observed a similar phase-dependent variation of PER2::LUC stability in CKO cells, with a smaller (± 20%) but significant difference between opposite phases of the oscillation (Fig 4D). To test if a 20% variation in protein half-life, in the absence of any underlying mRNA abundance rhythm, was sufficient to account for our experimental observations given the intrinsically high turnover of PER2, we made a simple mathematical model using experimentally derived values for mRNA level, protein half-life and translation (Figs 3C and EV4). We found that the model produced PER2::LUC levels that closely approximate our experimental observations (Fig 4 E). Thus whilst we cannot absolutely discount the possibility that rhythmic translation contributes to the PER2::LUC rhythms in CKO cells, we found no evidence to support this, whereas experimental observations and theoretical modelling do suggest rhythmic PER2 degradation alone is sufficient to explain the residual bioluminescence rhythms we observe in CKO PER2::LUC fibroblasts.

### CK1δ/ε and GSK3 contribute to CRY-independent PER2 oscillations

PER2 stability is primarily regulated through phosphorylation by casein kinases (CK) 1δ and 1ε, which phosphorylate PER2 at phosphodegron sites to target it for proteasomal degradation (Lee *et al*, 2009; Philpott *et al*, 2020). In this context, CK1δ/ε frequently operates in tandem with glycogen synthase kinase (GSK) 3α/β, as occurs in the regulation of β-catenin stability (O'Neill *et al*, 2013; Robertson *et al*, 2018). Interestingly, both CK1δ/ε and GSK3α/β have a conserved role in determining the speed at which the eukaryotic cellular circadian clock runs (Hastings *et al*, 2008; Causton *et al*, 2015), both in the presence and absence of transcription (Hirota *et al*, 2008; Meng *et al*, 2008; O'Neill *et al*, 2011; Beale *et al*, 2019). This is despite the fact that the clock proteins phosphorylated by these kinases are highly dissimilar between animals, plants and fungi (Causton *et al*, 2015; Wong & O'Neill, 2018).

We hypothesised that the PER2::LUC rhythm in CKO cells reflects the continued activity of a post-translational timekeeping mechanism that involves CK1δ/ε and GSK3α/β, which results in the differential phosphorylation and turnover of clock protein substrate effectors such as PER2 during each circadian cycle (O'Neill *et al*, 2013). To test this, we incubated WT and CKO cells with selective pharmacological inhibitors of CK1δ/ε (PF670462; PF) and GSK3α/β (CHIR99021; CHIR), which have previously been shown to slow down, and accelerate, respectively, the speed at which the cellular clock runs in a wide range of model organisms (Badura *et al*, 2007; Hirota *et al*, 2008; O'Neill *et al*, 2011; Causton *et al*, 2015). As a control we used KL001, a small molecule inhibitor of CRY degradation (Hirota *et al*, 2012), which has previously been shown to affect cellular rhythms in WT cells via increased CRY stability.

We found that inhibition of CK1δ/ε and GSK3-α/β had the same effect on circadian period in CKO cells, CPKO cells and WT controls (Figs 5A and B, and EV5A, B and D). In contrast, KL001 increased period length and reduced amplitude of PER2::LUC expression in WT cells but had no significant effect on post-translationally regulated PER2::LUC rhythms in CKO cells (Figs 5C and EV5C). Besides confirming the specific mode of action for KL001 in targeting CRY stability, these observations implicate CK1δ/ε and GSK3α/β in

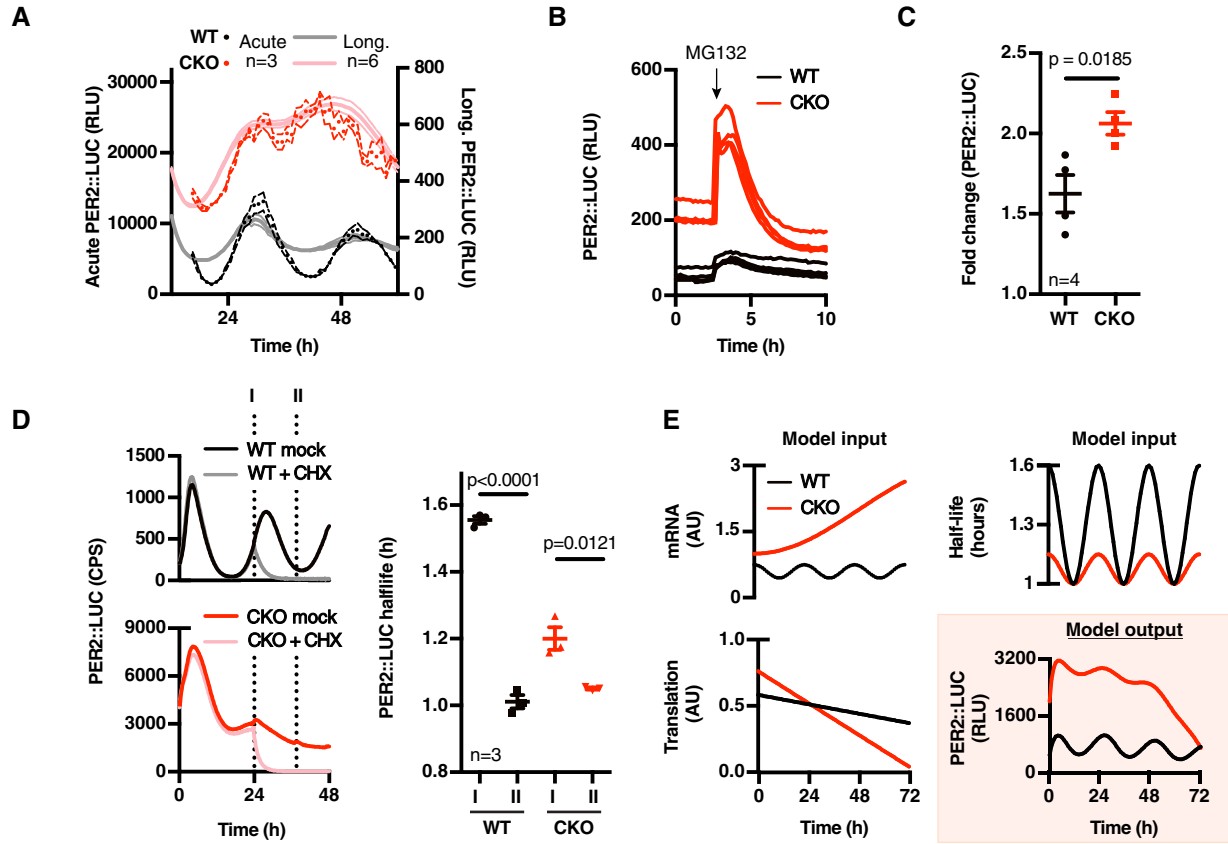

**Figure 4. PER2::LUC stability oscillates in CRY-deficient cells.**

A  Actual PER2::LUC levels (dark symbols (3-h moving average, $n = 3 \pm$ SEM, 4 outliers removed)) as assayed in acute luciferase assays on cell lysates from cells harvested every hour over 48 h, compared with parallel longitudinal co-recordings from cells in the presence of 0.1 mM luciferin (light lines ($n = 6$, mean $\pm$ SEM)).

B  PER2::LUC recording of asynchronous WT and CKO cells pulsed with proteasome inhibitor MG132 (10 μM, applied at the arrow) ($n = 3$, mean $\pm$ SEM).

C  Quantification of relative PER2::LUC induction upon proteasome inhibition, $n = 3$, mean (solid) $\pm$ SEM (dashed). *P*-value was calculated by unpaired *t*-test.

D  Phase-dependent PER2::LUC half-life was determined by inhibiting translation at different circadian phases and fitting the resulting data with a one-phase exponential decay curve ($n = 3$, mean $\pm$ SEM). Left image depicts the timing of cycloheximide (CHX, 10 μM) pulses (labelled I (PER2 levels going up) and II (PER2 levels going down)), plotted on PER2::LUC bioluminescence traces of control cells (dark colours). A representative trace of CHX-treated cells at time point I is shown in light colours. See Fig EV4D and F for more raw data and time points. Right image shows quantifications, and *P*-values were calculated by unpaired *t*-test.

E  A simple model incorporating mRNA, protein translation and PER2::LUC stability that were measured experimentally (inputs) shows that the observed oscillating stability of PER2 is sufficient to generate rhythmic PER2::LUC expression (output).

regulating the post-translational rhythm reported by PER2::LUC in CKO cells.

# Discussion

We found that CRY-mediated transcriptional feedback in the canonical TTFL clock model is dispensable for cell-autonomous circadian timekeeping in cellular models. Whilst we cannot exclude the possibility that in the SCN, but not fibroblasts, PER alone may be competent to effect transcriptional feedback repression in the absence of CRY, we are not aware of any evidence that would render this possibility biochemically feasible.

Circadian rhythms of PER abundance were observed in CKO SCN slices and fibroblasts, as well as Timeless-deficient flies, indicating that the post-translational mechanisms that normally confer circadian rhythmicity onto PER proteins in WT cells remain ostensibly intact in the absence of canonical transcriptional feedback repression. Importantly however, CKO PER2 rhythms were only observed in a minority of recordings (~ 30%), and when observed, they showed increased variance of period and sensitivity to perturbation. This reduced capacity to perform without failure under a wide range of conditions means that CRY-deficient oscillations are less robust than those in WT cells (Merriam-Webster Dictionary, 2020), and the reduced robustness of oscillations may explain or contribute to the obvious impairment of circadian physiological organisation at the organismal scale. We were unable to identify all of the variables that contribute to the apparent stochasticity of CKO PER2::LUC oscillations, and so cannot distinguish whether this variability arises from reduced fidelity of PER2::LUC as a circadian reporter or impaired timing function in CKO cells. In consequence, we restricted our study to those recordings in which clear bioluminescence rhythms were observed, enabling the interrogation of TTFL-independent cellular timekeeping.

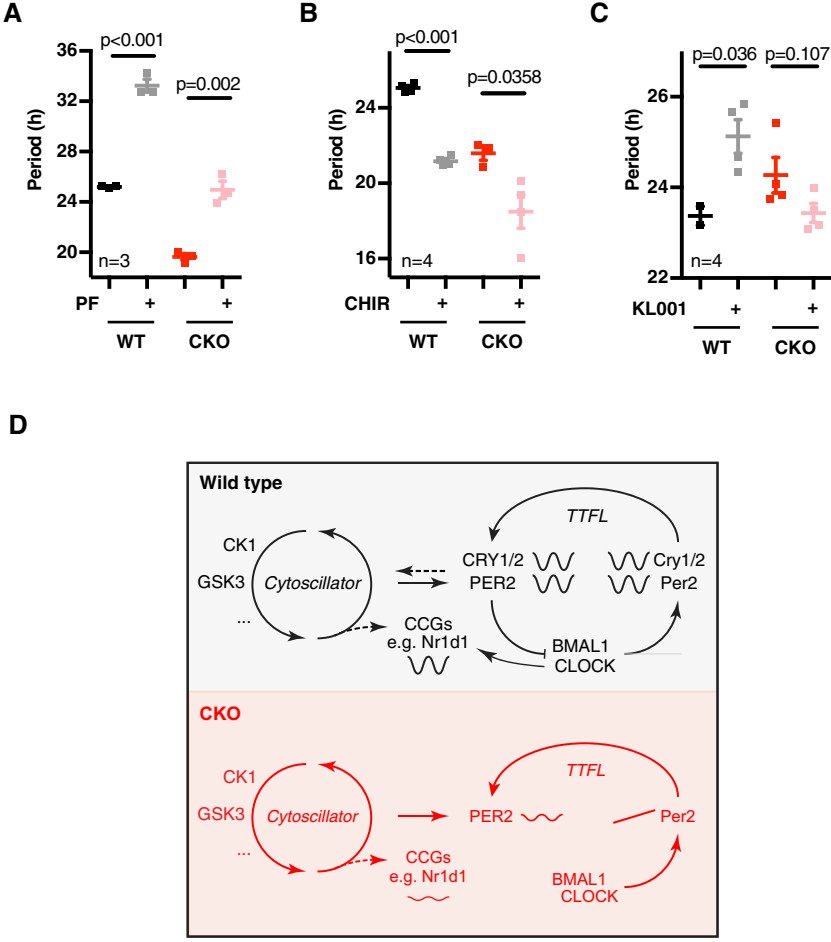

**Figure 5. A role for CK1 and GSK3 in the cytoplasmic oscillator.**

A Period (*n* = 3, mean ± SEM) analyses of WT and CKO PER2::LUC cells in the presence or absence of CK1δ/ε inhibitor PF670462 (0.3 μM; PF). *P*-values were calculated by unpaired *t*-test.

B As in (A, *n* = 4), GSK3 inhibitor CHIR99021 (5 μM; CHIR).

C As in (A, *n* = 4), in the presence of CRY inhibitor KL001 (1 μM).

D Schematic model integrating CRY-independent timekeeping into the existing canonical model of the circadian clock. The CRY-dependent gene expression feedback loop (TTFL) is required for most circadian regulation of transcriptional clock-controlled genes (CCGs) and therefore for robustness and behavioural and physiological rhythmicity. However, it is dispensable for circadian timekeeping *per se*, as reported by residual oscillations in PER2 protein levels, suggestive of the existence of a coupled underlying (cytosolic) timekeeping mechanism involving CK1 and GSK3 (cytoscillator).

Data information: See Fig EV5 for raw data.

In the field of chronobiology, CKO cells and mice are often used as clock-deficient models. Indeed, canonical circadian transcriptional output is essentially absent from these models (Hoyle *et al*, 2017; Ode *et al*, 2017), and thus for studying TTFL-mediated control of overt physiology, they are appropriate negative controls. However, as the underlying timekeeping mechanism seems at least partially intact, we consider it inappropriate to describe CKO cellular models as arrhythmic. Indeed, rest/activity behaviour of CKO mice does entrain to daily cycles of restricted feeding (Iijima *et al*, 2005), which is SCN-independent (Storch & Weitz, 2009). We also observed (about) daily rest–activity cycles *in vivo* and SCN PER2:: LUC rhythms *ex vivo* that, whilst being 20–30% shorter and less robust than WT controls, suggests CRY-independent timing mechanisms can co-ordinate communication between cells under some

conditions. Thus, non-TTFL-mediated timekeeping seems sufficient to serve as an almost daily interval timer in cells and *in vivo* (Crosby *et al*, 2019), but rhythmic precision, robustness and physiological function require the participation of CRY-dependent processes.

Previous studies have reported isolated CKO cells to be entirely arrhythmic (Sato *et al*, 2006; Ukai-Tadenuma *et al*, 2011; Ode *et al*, 2017), in stark contradiction with our findings (see also technical discussion). However, most such studies measured changes in transcription either by quantitative RT–PCR, or with luciferase fusions to fragments of the *Bmal1*, *Per* and *Cry* promoters which we also found to be arrhythmic in CKO cells. We did observe low amplitude oscillations in *Nr1d1* promoter activity, however. It may be pertinent to report that our MEF recordings only revealed circadian oscillations in *Nr1d1*-promoter activity, and only in bicarbonate-buffered

medium supplemented with 1 mM luciferin and 10% serum (Fig 3 D), but not in low serum or HEPES-buffered media, as employed in other studies that used different circadian reporters and may have employed sub-saturating concentrations of luciferin (Feeney *et al*, 2016a). It is also plausible that the high sensitivity of the electron-multiplying CCD camera we used for these bioluminescence assays allows the quantification of biological rhythms that were not detectable using other approaches (Crosby *et al*, 2017).

Although several mechanisms for circadian regulation of translation have been described (Jouffe *et al*, 2013; Lipton *et al*, 2015), we did not find any contribution of rhythmic translation to CRY-independent rhythms. In fact, the BMAL1-S6K1 interaction that mediates BMAL1's interaction with the translational apparatus is absent from CKO cells (Fig EV4C), implying a possible role for CRY proteins in this complex. Instead, we found an overt circadian regulation of PER2::LUC stability that persists in the absence of CRY proteins and which was sufficient to account for the observed PER2::LUC rhythms in a simple mathematical model. Persistent post-translational regulation of PER stability/activity may also account for the results of earlier overexpression studies, in mammalian cells and flies, where constitutive *Per* mRNA expression resulted in rhythmic PER protein abundance (Yang & Sehgal, 2001; Yamamoto *et al*, 2005; Fujimoto *et al*, 2006), whereas *Per* overexpression should really abolish rhythms if *Per* mRNA levels are the fundamental state variable of the oscillation. This interpretation has marked similarities with recent reports in the fungal clock model, *Neurospora crassa*, where experiments have suggested that post-translationally regulated cycles in the activity of the FRQ clock protein, not its turnover, are the critical determinant of downstream circadian gene regulation (Larrondo *et al*, 2015).

Indeed, our observations may not be particularly surprising when one considers that post-translational regulation of circadian timekeeping is ubiquitous in eukaryotes, with the period-determining function of CK1δ/ε and GSK3α/β being conserved between the animal, plant and fungal clocks (Lee *et al*, 2009; Hirota *et al*, 2010; O'Neill *et al*, 2011; Yao & Shafer, 2014; Causton *et al*, 2015; Wong & O'Neill, 2018), despite their clock protein targets being highly dissimilar between phylogenetic kingdoms. Importantly, we observed that pharmacological inhibition of these kinases elicited the same period-lengthening and period-shortening effects on CRY-independent rhythms as on WT rhythms. This has implications for our understanding of the role that these kinases play in the cellular clock mechanism, since in the absence of TTFL-mediated timekeeping their effects cannot be executed through regulation of any known transcriptional clock component.

Given similar findings across a range of model systems, including isolated red blood cells (Wong & O'Neill, 2018), the simplest interpretation of our findings entails an underlying, evolutionarily conserved post-translational timekeeping mechanism: a "cytoscillator" (Hastings *et al*, 2008) that involves CK1δ/ε and GSK3α/β, and can function independently of canonical clock proteins, but normally reciprocally regulates with cycles of clock protein activity through changes in gene expression (Qin *et al*, 2015). This cytoscillator confers 24-h periodicity upon the activity and stability of PER2, and most likely to other clock protein transcription factors as well (Fig 3D). However, a purely post-translational timing mechanism should be rather sensitive to environmental perturbations and biological noise (Ladbury & Arold, 2012), as seen for CKO cells. Due to the geometric nature of their underlying oscillatory mechanism, relaxation oscillators are known to be particularly insensitive to external perturbations and are prevalent in noisy biological systems (Muratov & Vanden-Eijnden, 2008). We therefore suggest that in WT cells, low amplitude, cytoscillator-driven circadian cycles of clock protein activity are coupled with, reinforced and amplified by a damped TTFL-based relaxation oscillation of stochastic frequency (Chickarmane *et al*, 2007), resulting in high-amplitude, sustained circadian rhythms in both clock and clock-controlled gene expression. Indeed, mathematical modelling shows that such coupling can both drive the emergence of sustained oscillations in overdamped systems (In *et al*, 2003) and play an important role in maintaining robust oscillations in a random environment (Medvedev, 2010). This model is consistent with recent observations in the clocks of the prokaryotic cyanobacterium *Synechoccocus elongatus* (Qin *et al*, 2010; Teng *et al*, 2013) as well as the fungus *Neurospora crassa* (Larrondo *et al*, 2015), and the alga *Ostreococcus tauri* (Feeney *et al*, 2016b; see Appendix for an extended discussion).

This model is attractive for several reasons. First, it may explain the discrepancy between SCN and behavioural studies in CKO mice, in that residual timekeeping can be observed in cultured SCN PER2::LUC activity, whereas behavioural rhythmicity is not observed in constant darkness following standard 12h:12h light:dark entrainment, but is expressed under specific non-standard conditions (Iijima *et al*, 2005; Ono *et al*, 2013b). Considering the CKO cellular clock's shorter intrinsic period, as well as the profound robustness conferred upon SCN timekeeping by interneuronal coupling (Yamaguchi *et al*, 2003; Welsh *et al*, 2010;), it seems plausible that 24-h cycles may simply lie outside the range of circadian entrainment for CKO SCN *in vivo*, similar to the *tau* mutant hamster and humans with familial advanced sleep phase syndrome (Ptáček *et al*, 2007; Meng *et al*, 2008). Whereas *ex vivo*, or following the strong synchronising cue imposed by transition from constant light to constant darkness (Chen *et al*, 2008), non-transcriptional cellular mechanisms are sufficient to impart circadian regulation to CKO neuronal activity that is amplified by neuropeptidergic Ca$^{2+}$/cAMP-signalling (O'Neill & Reddy, 2012), facilitating the same temporal consolidation of locomotor activity observed in wild-type mice but with shorter period and less precision.

Second, whilst the evidence is indisputable that transcriptional feedback repression is critical for circadian co-ordination of global gene expression, physiology and behaviour, the evidence that these regulatory gene expression circuits are inherently possessed of approximately 24-h rhythmicity is weak (reviewed in Lakin-Thomas, 2006; Putker & O'Neill, 2016; Wong & O'Neill, 2018). Post-translational regulation of clock protein stability, activity and localisation, however, is already well established as the primary determinant of the delay constants that allow the oscillation to persist with a period of about 1 day in all studied eukaryotic cells (Gallego & Virshup, 2007; van Ooijen *et al*, 2011; Top *et al*, 2018; Wong & O'Neill, 2018). We simply suggest that transcriptional feedback repression is not essential for circadian timekeeping *per se*, but amplifies the rhythms to increase robustness *via* hysteresis, when engaged, and also to confer tissue and cell-type-specific functionality (Wong & O'Neill, 2018). Our paradigm here being the cell division cycle, where the essential timing mechanism is also post-translational, and persists in enucleated cells (Hara *et al*, 1980; Pomerening *et al*, 2005).

Third, there is no evidence that TTFL-mediated oscillations would not damp to a steady state without post-translational input (Wong & O'Neill, 2018). In contrast, there are several examples in the eukaryotic lineage, where circadian timekeeping persists in the absence of cycling gene expression (Lakin-Thomas, 2006; Sweeney & Haxo, 1961; O'Neill & Reddy, 2011; O'Neill *et al*, 2011). For example, the period of circadian rhythms in human cells and *Ostreococcus tauri* is regulated by CK1, both in the presence and absence of nascent transcription (O'Neill *et al*, 2011; Beale *et al*, 2019) similar to the rhythm reported by PER2::LUC in CKO cells we report here.

Interestingly, the concept of the eukaryotic post-translational clock mechanism we propose is not new (Roenneberg & Merrow, 1998; Merrow *et al*, 2006; Qin *et al*, 2010; Jolley *et al*, 2012) and resembles the KaiA/B/C mechanism elucidated in cyanobacteria (Nakajima *et al*, 2005; Teng *et al*, 2013). The challenge will now be to identify additional factors that, in concert with CK1 and GSK3, and protein phosphatase 1 (Lee *et al*, 2011), serve as the functional equivalents of KaiA/B/C, allowing reconstitution of the mammalian circadian clock *in vitro* (Nakajima *et al*, 2005; Millius *et al*, 2019).

Here we have uncovered PER2 as a node of interaction between a putative cytoscillator mechanism and the canonical circadian TTFL (Fig 5D). It is unlikely however that PER2 is the only interaction between the two, as *Per2*[-/-] knockout cells and mice exhibit competent circadian timekeeping (Xu *et al*, 2007), suggesting redundancy in this respect. Indeed, the residual noisy but rhythmic activity of the *Nr1d1*-promoter in the absence of both PER1/2 and CRY1/2 (Fig 3D) suggests another point of connection between the cytoscillator and TTFL. Moreover, both CK1 and GSK have been implicated in the phosphorylation and regulation of many other clock proteins (see Table S2 in Causton *et al*, 2015, also reviewed in O'Neill *et al*, 2013). Some or all of these targets may play a role in coupling the cytoscillator with TTFL-mediated clock output. We believe that it is now imperative to delineate the specific means by which the TTFL couples with the cytoscillator to effect changes in circadian phase in order that the two resonate with a common frequency.

## Conclusion

Whilst the contribution of clock protein transcription factors to the temporal co-ordination of gene expression, physiology and behaviour is unambiguous, the primacy of transcriptional feedback repression as the ultimate arbiter of circadian periodicity within eukaryotic cells is not. Similar to the conserved kinase-dependent regulation of the cell division cycle, we suggest the circadian cycle in diverse eukaryotes is conserved from a common ancestor, with diverse TTFL components having been recruited throughout speciation to impart robustness, signal amplification and functional specificity to the oscillation.

# Materials and Methods

Reagents were obtained from Sigma unless stated otherwise. More detailed Materials and Methods can be found in the Appendix.

## Mouse work

All animal work was licensed under the UK Animals (Scientific Procedures) Act 1986, with Local Ethical Review by the Medical Research Council. *Cry1/2*-null mice were kindly provided by G. T. van der Horst (Erasmus MC, Rotterdam, The Netherlands) (Horst & Muijtjens, 1999), PER2::LUC mice by J. S. Takahashi (UT Southwestern, USA) (Yoo *et al*, 2004) and *Cry1*:LUC mice by M. Hastings (MRC LMB, Cambridge, UK) (Maywood *et al*, 2013). All lines were maintained on a C57BL/6J background. For mouse behavioural studies, two independent recordings were made from male and female CKO PER2::LUC aged 2–5 months, with age- and gender-matched PER2::LUC controls, singly housed in running wheel cages with circadian cabinets (Actimetrics). They were then subject to 7 days 12 h:12 h LD cycles or 7 days constant light (400 lux), and then maintained in constant darkness with weekly water and food changes. Locomotor activity was recorded using running wheel activity and passive infrared detection, which was analysed using the periodogram function of ClockLab (Actimetrics) with a significance threshold of $P = 0.0001$. SCN organotypic slices from 7- to 10-day-old pups were prepared as previously described (Hastings *et al*, 2005), and bioluminescence recorded using photomultiplier tubes (Hamamatsu).

## Mammalian cell culture

Primary fibroblasts were isolated from lung tissue (Seluanov *et al*, 2010) of adult wild-type (WT) and *Cry1*[-/-],*Cry2*[-/-] (CKO) PER2::LUC male and female mice, and WT and CKO *Cry1*:LUC mice. Stable WT, CKO and *Cry1*[-/-],*Cry2*[-/-], *Per1*[-/-], *Per2*[-/-] (CPKO) mouse embryonic fibroblasts (MEFs) expressing transcriptional luciferase reporters for clock gene activity were generated by puromycin selection and cultured as described previously (Valekunja *et al*, 2013). MEFs were seeded into 96-well white plates at $10^4$ cells/well and grown to confluency for 5 days under temperature cycles (12 h:12 h, 32°C:37°C) to synchronise circadian rhythms. Primary fibroblasts were cultured as described previously (O'Neill & Hastings, 2008) and immortalised by serial passage (Xu, 2005). CRY deficiency was confirmed by PCR (see Appendix) and Western blotting [guinea pig-anti-CRY1 and CRY2 antibodies (Lamia *et al*, 2011)]. NIH3T3 fibroblasts expressing SV40::LUC have been described before (Feeney *et al*, 2016a).

## Luciferase recordings

Fibroblast recordings were performed in air medium (either HEPES or MOPS buffered (20 mM), either in airtight sealed dishes (in non-humidified conditions) or open in humidified conditions (0% $CO_2$). Air medium stock was prepared as described previously (O'Neill & Hastings, 2008) and supplemented with 2% B-27 (Life Technologies, 50×), 1 mM luciferin (Biosynth AG), 1× glutamax (Life Technologies), 100 units/ml penicillin/100 μg/ml streptomycin and 1% FetalClone™ III serum (HyClone™). Final osmolarity was adjusted to 350 mOsm with NaCl. Recordings were preceded by appropriate synchronisation (see Appendix for details) in the presence of 0.3 mM luciferin to prevent artificially high bioluminescence activity at the start of the recording, and started immediately after a medium change from culture medium

into air medium. The presented MEF recordings were performed in an ALLIGATOR (Crosby *et al*, 2017) and employed bicarbonate-buffered Dulbecco's modified Eagle medium (10569010) with penicillin/streptomycin and 1 mM luciferin in a humidified incubator at 5% $CO_2$, also supplemented with 2% B-27 and 10% FetalClone™ III serum. A range of other media conditions were explored but did not produce detectable bioluminescence rhythms in CKO or CPKO cells (not shown). For pharmacological perturbation experiments (unless stated otherwise in the text), cells were changed into drug-containing air medium from the start of the recording. Mock treatments were carried out with DMSO or ethanol as appropriate.

Bioluminescence recordings were performed in a lumicycle (Actimetrics), a LB962 plate reader (Berthold technologies) or an ALLIGATOR (Cairn Research). Acute luciferase assays were performed using a Spark 10 M microplate reader (Tecan).

## Biochemistry

The number of PER2 molecules was determined by harvesting a known number of synchronised WT and CKO cells at the peak of PER2 expression and comparing the Luciferase activity to a standard curve of recombinant luciferase (see Appendix for details). Three technical replicates were measured in every experiment, and the experiment was carried out three times. A representative experiment is shown.

For determining *Per2*::Luc and *Bmal1* mRNA levels, synchronised cells were harvested from constant conditions in triplicate every 4 h from 24 h up to 48 h after media change. RNA extraction and qPCR were performed as detailed in the Appendix. Analysis involved three technical and three biological replicates. Relative amounts of mRNA were determined by comparing the samples to a standard curve and expressed relatively to ribosomal RNA Rns18s.

For comparing longitudinal PER2::LUC recordings to the actual PER2::LUC protein levels (longitudinal vs. acute luciferase assays), synchronised WT and CKO cells (cultured in absence of luciferin) were harvested every hour (in triplicate) from 16 h up to 64 h after media change, whilst co-cultures were recorded for bioluminescence in the presence of luciferin. Luciferase activity in acute assays was determined as detailed in Appendix.

For assaying the interaction between BMAL1 and PER2::LUC, synchronised cells were harvested directly from temperature cycles at the expected peak of PER2::LUC expression (4 h after change to 32°C) and BMAL1 was precipitated as described in Appendix. PER2::LUC co-immunoprecipitation was measured in a luciferase assay by mixing the BMAL1-loaded beads in luciferase assay buffer (15 mM $MgSO_4$, 30 mM HEPES, 300 μM luciferin, 1 mM ATP, 10 mM 2-mercaptoethanol) and measuring luciferase activity in a Berthold plate reader. The results were corrected for input and plotted relatively to the WT IgG pulldown.

To study the interaction of BMAL1 with S6K and eIF4, cells were synchronised by a 2-h dexamethasone pulse, after which they were changed into normal growth medium. 12 and 24 h after the medium change, BMAL1 immunoprecipitation was executed as described in Appendix. Samples were analysed by Western blot for the presence of BMAL1, S6K and eIF4 (Cell Signaling, resp. #2708 and #2013).

### *Drosophila* experiments

All fly strains were kept in standard cornmeal food under 12 h:12 h LD cycles at constant 25°C (LD cycles). The following control strains were included in the experiments: *per[01]*, Canton S and *w[1118]*. The generation of *Tim[Out]* flies, crossings with XLG-luc flies (Veleri *et al*, 2003) and details of recordings are described in Appendix. In short, 3- to 7-day-old flies were entrained for 3 days LD cycles before being loaded individually into the wells of a microtiter plate containing the food-luciferin substrate (15 mM luciferin). Recordings were performed under constant darkness at 26°C over 7 days. Bioluminescence from each fly was background subtracted, summed into 2-h bins and then detrended using a 24-h moving average. Rhythmicity of averaged traces was tested by least-square fitting, comparing a circadian damped sine wave with the null hypothesis (straight line), as described below.

### Analysis

All analyses were performed in GraphPad Prism versions 7 and 8. Where indicated, data were detrended using moving average subtraction, where temporal window of the moving average was refined iteratively until it matched with the period of oscillation derived as follows. Period analysis was performed either manually, or by least-square fitting to a circadian damped sine wave with a linear baseline:

$$y = (mx + c) + a \exp^{-kx} \sin\left(\frac{2\pi x - r}{p}\right)$$

where $m$ is the gradient of the baseline, $c$ is the $y$ offset, $k$ describes the rate of dampening, $a$ the amplitude, $r$ the phase and $p$ the period. Reported *P*-values for the curve fit are those produced by the comparison of fits function in Prism 8, where the null hypothesis was a straight line ($y = mx + c$), i.e. change over time but with no oscillatory component. The simpler model was preferred unless the sine wave fit produced a better fit with $P < 0.05$.

For the mathematical model in 4E, we assumed that PER2::LUC translation at time ($t$) is a function of *Per2*::*Luc* mRNA abundance, corrected for the changes we observed for global translation rate over time; and that PER2::LUC degradation rate follows one-phase exponential decay kinetics where the decay constant is defined by a sine wave with 24-h periodicity, with the amplitude, phase and other parameters being derived entirely from experimental measurements. See Appendix for details.

## Data availability

This study includes no data deposited in external repositories.

**Expanded View** for this article is available online.

## Acknowledgements

We thank biomedical technical staff at Medical Research Council (MRC) Ares facility and LMB facilities for assistance, G.T. van der Horst and J.S. Takahashi for sharing rodent models, M.H. Hastings and E.S. Maywood for providing

reagents and input, K. Lamia for providing reagents, and P. Crosby, D.S. Tourigny, J.E.C. Jepson, C.P. Kyriacou and H.R. Pelham for valuable discussion. MP was supported by the Dutch Cancer Foundation (KWF, BUIT-2014-6637) and EMBO (ALTF-654-2014). JON was supported by the Medical Research Council (MC_UP_1201/4) and the Wellcome Trust (093734/Z/10/Z). NP and RF were supported by the Deutsche Forschungsgemeinschaft FKZ (Pe1798/2-1). AS and CPS were supported by the National Institutes of Health (GM118102). NMR was supported by the Medical Research Council (MR/S022023/1).

## Author contributions

MP and JSON designed the study, analysed the data and wrote the manuscript. AZ and NMR performed mouse behavioural studies. MP, DCSW, ES, NPH, KAF, MVO, RSE and JSON performed cell experiments. CPS and AS generated MEF cell lines. MDE performed SCN experiments. K-FC, RF, NP and JSON performed fly experiments. JEC performed tissue collection and husbandry. All authors commented on the manuscript.

## Conflict of interest

The authors declare that they have no conflict of interest.

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
