## [Review Process File · The EMBO Journal]

CRYPTOCHROMES confer robustness, not rhythmicity, to circadian timekeeping

Marrit Putker, David Wong, Estere Seinkmane, Nina Rzechorzek, Aiwei Zeng, Nathaniel Hoyle, Johanna Chesham, Mathew Edwards, Kevin Feeney, Robin Fischer, Nicolai Peschel, Ko-Fan Chen, Michael Vanden Oever, Rachel Edgar, Christopher Selby, Aziz Sancar, and John O'Neill
DOI: 10.15252/embj.2020106745

Corresponding author(s): John O'Neill (oneillj@mrc-lmb.cam.ac.uk)

Review Timeline:	Transfer from Review Commons:	9th Sep 20
	Editorial Decision:	29th Sep 20
	Revision Received:	28th Oct 20
	Editorial Decision:	16th Nov 20
	Revision Received:	8th Dec 20
	Accepted:	18th Dec 20

Editor: Ieva Gailite

Review
COMMONS

Transaction Report: This manuscript was transferred to The EMBO Journal following peer review at Review Commons.

Revision 0

Review #1

1. How much time do you estimate the authors will need to complete the suggested revisions:

Estimated time to Complete Revisions (Required)Rev

(Decision Recommendation)

Between 1 and 3 months

2. Evidence, reproducibility and clarity:

Evidence, reproducibility and clarity (Required)

****Summary:**** This interesting study by Putker et al. showed that circadian rhythmicity persists in several typical circadian assay systems lacking Cry, including Cry knockout mouse behavior and gene expression in Cry knockout fibroblasts. They further demonstrated weak but significant circadian rhythmicity in Cry- and Per- knockout cells. Cry- (and potentially Per-)-independent oscillations are temperature compensated, and CKId/e still has a role in the period regulation of Cry-independent oscillations. ****Major comments:**** 1) The authors propose that the essential role of mammalian Cryptochrome is to bring the robust oscillation. As the authors analyze in many parts, the robustness of oscillation can be validated by the (relative) amplitude and phase/period variation, both of which should be affected significantly by the method for cell synchronization. Unfortunately, the method for synchronization is not adequately written in this version of supplementary information. This reviewer has no objection to the "iterative refinement of the synchronization protocol" but at least the correspondence between which methods were used in which experiments needs to be clearly explained. The detailed method may be found in the thesis of Dr. Wong, but the methods used in this manuscript need to be detailed within this manuscript. 2) The authors revealed that CKO mice have apparent behavioral rhythmicity under the condition of LL>DD. This is an intriguing finding. However, it should be carefully evaluated whether this rhythmicity (16 hr cycle) is the direct consequence of circadian rhythmicity observed in CKO and CPKO cells (24 hr cycle) because the period length is much different. Is it possible to induce the 16 hr periodicity in CKO mice behavior by 16 hr-L:16 hr-D cycle? Would it be a plausible another possibility that the 16 hr rhythmicity is the mice version of internal desynchronization or another type of methamphetamine-induced-oscillation/food-entrainable-oscillation? 3) The authors proposed that CKId/e at least in part is the component of cytosillator (Fig. 5D), and turnover control of PER (likely to be controlled by CKId/e) may be an interaction point between cytosillator and canonical circadian TTFL (Fig. 4). Strictly speaking, this model is not directly supported by the experimental setting of the current

manuscript. The contribution of CKId/e is evaluated in the presence of PER by monitoring the canonical TTFL output (i.e. PER2::LUC); thus it is not clear whether the kinase determines the period of cytos oscillator. It would be valuable to ask whether the PF and CHIR have the period-lengthening effect on the Nrd1:LUC in the CPKO cell. ****Minor comments:**** 4) The authors argue that the CKO cells' rhythmicity is entrained by the temperature cycle (Fig. 2C). Because the data of CKO cell only shows one peak after the release of constant temperature phase, it is difficult to conclude whether the cell is entrained or just respond to the final temperature shift. 5) It would be useful for readers to provide information on the known phenotype of TIMELESS knockout flies; TIM is widely accepted as an essential component of the circadian clock in flies; are there any studies showing the presence of circadian rhythmicity in Tim-knockout flies (even if it is an oscillation seen in limited conditions, such as the neonatal SCN rhythm in mammalian Cry knockout)? 5) Figure 3C shows that the amount of PER2::LUC mRNA changes ~2 fold between time = 0 hr and 24 hr in the CKO cell. This amplitude is similar to that observed in WT cell although the peak phase is different. Does the PER2::LUC mRNA level show the oscillation in CKO cells? 6) Figure 3D: the authors discuss the amplitude and variation (whether the signal is noisier or not) of reporter luciferase expression between different cell lines. However, a huge difference in the luciferase signal can be observed even in the detrended bioluminescence plot. This reviewer concerns that some of the phenotypes of CKO and CPKO MEF reflect the lower transfection efficiency of the reporter gene, not the nature of circadian oscillators of these cell lines.

3. Significance:

Significance (Required)

Although Cryptochrome (Cry) has been considered a central component of the mammalian circadian clock, several studies have shown that circadian rhythms are maintained in the absence of Cry, including in the neonate SCN and red blood cells. Thus, although the need for Cry as a circadian oscillator has been debated, its essential role as a circadian oscillator remains established, at least in the cell-autonomous clock driven by the TTFL. This study provides additional evidence that the circadian rhythmicity can persist in the absence of Cry. More general context, the presence of a non-TTFL circadian oscillator has been one of the major topics in the field of circadian clocks except for the cyanobacteria. In mammals, the authors' and other groups lead the finding of circadian oscillation in the absence of canonical TTFL by showing the redox cycle in red blood cells (O'Neil, Nature 2011). The presence of circadian oscillation in the absence of Bmal1 is also reported recently (Ray, Science 2020). Bmal1(-CLOCK), CRY, and PER compose the core mechanism of canonical circadian TTFL; thus, this manuscript put another layer of evidence for the non-TTFL circadian oscillation in mammals. Overall, the manuscript reports several surprising results that will receive considerable attention from the circadian community. This reviewer has expertise in the field of mammalian circadian clocks, including genomics, biochemistry, and mice's behavior analysis.

Review #2

1. How much time do you estimate the authors will need to

complete the suggested revisions:

Estimated time to Complete Revisions (Required)

(Decision Recommendation)

Between 1 and 3 months

2. Evidence, reproducibility and clarity:

Evidence, reproducibility and clarity (Required)

In the canonical model of the mammalian circadian system, transcription factors, BMAL1/CLOCK, drive transcription of *Cry* and *Per* genes and CRY and PER proteins repress the BMAL1/CLOCK activity to close the feedback loop in a circadian cycle. The dominant opinion was that CRY1 and CRY2 are essential repressors of the mammalian circadian system. However, this was challenged by persistent bioluminescence rhythms observed in SCN slices derived from *Cry*-null mice (Maywood et al., 2011 PNAS) and then by persistent behavior rhythms shown by the *Cry1* and *Cry2* double knockout mice if they are synchronized under constant light prior to free running in the dark (Ono et al., 2013 PLOS One). In the manuscript, the authors first confirmed behavioral and molecular rhythms in the *Cry1/Cry2*-deficient mice and then provided evidence to suggest the rhythms of *Per2:LUC* and *Nr1d1:LUC* in CKOs are generated from the cytoplasmic oscillator instead of the well-studied transcription and translation feedback loop: Constant *Per2* transcription driven by BMAL1/CLOCK plus rhythmic degradation of the PER protein result in a rhythmic PER2 level in the absence of both *Cry1* and *Cry2*, which suggests a connection between the classic transcription- and translation-based negative feedback loops and non-canonical oscillators.

****Major points:**** Line 38-39, "Challenging this interpretation, however, we find evidence for persistent circadian rhythms in mouse behavior and cellular PER2 levels when CRY is absent." The rhythmic behavioral phenotype of *cry1* and *cry2* double knockout mice was first documented by Ono et al., 2013 PLOS ONE, in which eight *cry1* and *cry2* double knockout mice after synchronization in the light displayed circadian periods with different lengths and qualities. The paper reported two period lengths from the *Cry* mutant mice: "An eye-fitted regression line revealed that the mean shorter period was 22.86±0.4 h (n= 8) and the mean longer period was 24.66±0.2 h (n =9). The difference of two periods was statistically significant (p, 0.01).", either of which is quite different from the ~16.5 hr period in Figure 1B of the manuscript. A brief discussion on the period difference between studies will be helpful for readers to understand. Period information from the individual mouse should be calculated and shown since big period variations exist among CKO mice (Ono et al., 2013 PLOS One). The behavioral phenotype of *Cry*-null mice and luminescence from their SCNs are robustly rhythmic while fibroblasts derived from these mice only produce rhythms with very low amplitudes compared with those in WT, which may reflect the difference between the SCN's rhythm and peripheral clocks. The behavioral phenotype is supposed to be controlled mainly by SCN. However, most molecular analyses in the work were done with MEF and lung fibroblasts. These tissues may not be the best representative of the behavioral phenotype of the CKO mice. Stronger evidence is needed to fully exclude the possibility that in CKO cells, the rhythm is not generated by PERs' compensation for the loss of *Crys* to repress BMAL1 and CLOCK. Since the rhythms of *Per:LUC* or *Nr1d1:LUC* (Figures 3D and S3E) are much

weaker than those in WT, molecular analyses might not be sensitive enough to reflect the changes across a circadian cycle in the CKOs if the TTFL still occurs. CLOCK Δ 19 mutant mice have a ~4 hr longer period than WT (Antoch et al., 1997 Cell; King et al., 1997 Cell). CLOCK Δ 19; CKO cells or mice should be very helpful to address the question. Periods of Per:LUC and Nr1d1:LUC from the CLOCK Δ 19; CKO should be similar to those in the CKO alone if the transcription feedback does not contribute to their oscillations. Lines 51-52, "PER/CRY-mediated negative feedback is dispensable for mammalian circadian timekeeping" and lines 310-311, "We found that transcriptional feedback in the canonical TTFL clock model is dispensable for cell-autonomous circadian timekeeping in animal and cellular models." The authors have not excluded the possibility that the rhythmic behaviors of the CKO mice are derived from the PERs' compensation for the role of Crys in the feedback loop of the circadian clock in the SCN. In the fibroblasts, only two genes, Per2 and Nr1d1, have been studied in the work, which cannot be simply expanded to the thousands of circadian controlled genes. Also amplitudes of PER2:LUC and NR1D1:LUC in the CKOs are much lower than those in WT and no evidence has been provided to show that their weak rhythms are biologically relevant. **Minor points:** Lines 66-67, "... (Dunlap, 1999; Reppert and Weaver, 2002; Takahashi, 2016)." to "... (reviewed in Dunlap, 1999; Reppert and Weaver, 2002; Takahashi, 2016)." Line 70, "... (Liu et al., 2008..." to "... (Liu et al., 2008..." Lines 174-175, "Considering recent reports that transcriptional feedback repression is not absolutely required for circadian rhythms in the activity of FRQ..." Larrondo et al., 2015 paper says "however, in such Δ frq-1 cells, the amount of FRQ still oscillated, the result of cyclic transcription of frq and reinitiation of FRQ synthesis." The point of the paper is "we unveiled an unexpected uncoupling between negative element half-life and circadian period determination." instead of "...transcriptional feedback repression is not absolutely required for circadian rhythms in the activity of FRQ," Lines 249-252, "CKO cells exhibit no rhythm in Per2 mRNA (Figure 3C, D), nor do they show a rhythm in global translational rate (Figure S4A, B), nor did we observe any interaction between BMAL1 and S6K/eIF4 as occurs in WT cells (Lipton et al, 2015) (Figure S4C)." In figures 3D and S3E, in CKO and CPKO cells the Per2:LUC data without fitting look better than that of Nr1d1:LUC. But the Nr1d1:LUC rhythm became clear after fitting the raw data. So to better visualize the low amplitude rhythm, if any, of Per2:LUC and compare with Nr1d1:LUC, fitted the Per2:LUC data in CKOs and CPKOs in Figure 3D and S3E should be shown as what has been done to Nr1d1:LUC. Lines 258-259, "much less than the half-life of luciferase expressed in fibroblasts under a constitutive promoter" In figure S4D, the y-axis of the PER2::LUC is ~800 while the y-axis of the SV40::LUC is ~600000. The over-expressed LUC by the SV40 promoter might saturate the degradation system in the cell so the comparison is not fair. A weaker promoter with the level similar to Per2 should be used to make the comparison. Line 430, "sigma" to "Sigma". In figure S2, the classification of rhythms in Drosophila is not clear since even the "Robustly rhythmic" ones have high background noise. Detrending or fitting the data might be able to improve the quality of the rhythms prior to classification. In figure S3B, the original blots for Per2 including Input and IP should be shown. Supplemental information Line 44, "... (reviewed in (Lakin-Thomas,..." to "... (reviewed in Lakin-Thomas,..." Line 188, "Period CDS", the full name of CDS should be provided the first time it appears.

3. Significance:

Significance (Required)

The work suggests a link between the TTFL and non-canonical oscillators, which should be interesting to the circadian field.

Review #3

1. How much time do you estimate the authors will need to complete the suggested revisions:

Estimated time to Complete Revisions (Required)

(Decision Recommendation)

Less than 1 month

2. Evidence, reproducibility and clarity:

Evidence, reproducibility and clarity (Required)

****Summary:**** The paper "CRYPTOCHROMES confer robustness, not rhythmicity, to circadian timekeeping" by Putker et al. answers the question of whether or not the rhythmic abundance of clock proteins is a prerequisite for circadian timekeeping. They addressed this by monitoring PER2::LUC rhythms in WT and CRY KO (CKO) cells. CRY forms a complex with PER, which in turn represses the ability of CLOCK/BMAL1 to drive the expression of clock-controlled genes, including PER and CRY. Consistent with previous observations, the authors found residual PER2::LUC rhythms in CKO SCN slices, fibroblasts and in a functional analogue KO of CRY in Drosophila, even in the absence of rhythmic Per2 transcription due to the loss of CRY as a negative regulator of the oscillation. They have shown that these rhythms, in the absence of CRY, follow the formal definition of circadian rhythms. They attributed these residual PER2::LUC rhythms to the maintenance of oscillation in PER2::LUC stability independent of CRY, by testing the decay kinetics of luciferase activity when translation is inhibited. Moreover, they implicated the kinases CK1 [?] [?] [?] [?] and GSK3 to be involved in regulating PER2::LUC post-translational rhythms through kinase inhibitor studies. They concluded that CRY is not necessary for maintaining PER2::LUC rhythms, but plays an important role in reinforcing high-amplitude rhythms when coupled to a proposed "ctyoscillator" likely composed of CK1 [?] [?] [?] [?] and GSK3. ****Major comments:**** The authors have shown sufficient data that under different testing conditions (mice locomotor activity, SCN preps or fibroblasts), behavioral rhythms and PER2::LUC rhythms are still observed in the CRY KO (CKO) cells, contrary to a previous study (Liu et al., 2007). They also indicated limitations to some of the experimental work. However, there are some parts of the paper that need clarification to support their conclusions. 1. In Fig. 1A, the x-axes of the actograms for WT and CKO are different. While they mentioned this in the figure legend, and described the axis transformation in Fig. S1A, they need a justification statement about why they did this in the results. 2. In an attempt to show conservation of their proposed role for CRY, they tested the model system Drosophila melanogaster where TIMELESS serves as the functional analogue of CRY. While they

showed in the figures and described in the text that rhythms still persisted with lower relative amplitude in the TIMELESS-deficient flies, they did not describe any period differences between WT and mutant. Showing the period quantification in Supp. Fig. S2 using the robustly rhythmic datasets, and describing this data in the text, will strengthen their claim. In Fig. S2B, there is no clear distinction between the representative datasets shown for poorly rhythmic and arrhythmic, i.e. they all appear arrhythmic, without an indicated statistical test. The authors could present better representative data to better reflect the categories. 3. In Fig. 2A, the authors note the lack of rhythmicity in the CKO fibroblasts in the 1st three days at 37°C. How are the conditions here different from fibroblasts in Fig. 1E, where rhythms are seen during the 1st three days in CKO fibroblasts? 4. The authors claimed in the results section- "in contrast and as expected, Per2 mRNA in WT cells varied in phase with co-recorded PER2::LUC oscillations." but Fig. 3C does not show this expected lag between mRNA and protein levels. This needs to be explained 5. In Figs. 5A-B, the PER2::LUC periods in the CKO untreated cells seem to vary significantly between A, B, and C. While this could be due to the high variability in the rhythms that were previously described by the authors, the average periods here seem to be longer than the one reported in Fig. 1F. Are there specific condition differences? *Would additional experiments be essential to support the claims of the paper?*

1. There is sufficient experimental data to support the major claims; however some suggested experiments are listed below. a. If CKO exhibits residual rhythms in PER2::LUC, it would be interesting to know how CRY overexpression influences PER2::LUC rhythms, or point to previous reference papers which may have already shown such effects. The prediction would be PER2::LUC levels will still be rhythmic when CRY is overexpressed. What would be the extent of "robustness" conferred by CRY on PER2::LUC rhythms based on CRY KO and overexpression studies? b. The authors found that CK1 [?] [?] [?] [?] and GSK3 contribute to CRY-independent PER2 oscillations by showing that addition of kinase inhibitors affect the PER2::LUC period lengths in WT and CKO in the same manner. It would be interesting to know if a) PER2::LUC stability and b) PER2 phosphorylation status, is affected in WT and CKO in the presence of the inhibitors, or point to previous reference papers which may already have shown such effects. *Are the data and the methods presented in such a way that they can be reproduced?*

1. The protocol for the inhibitor treatments are not in the main or supplemental methods. *Are the experiments adequately replicated and statistical analysis adequate?*

1. All experiments had the sufficient number of technical and biological replicates to make valid statistical analyses. For Fig. S2, the authors used RAIN to assess rhythmicity in WT and mutant flies, but it is not clear whether the different categories (rhythmic, poorly rhythmic, and arrhythmic) were based on amplitude differences alone, or a combination of amplitude and p-values as determined by RAIN. **Minor comments:**

1. Are prior studies referenced appropriately?

Authors may wish to include Fan et al., 2007, Current Biology which demonstrated that cycling of CRY1, CRY2, and BMAL1 is not necessary for circadian-clock function in fibroblasts. *2. Are the text and figures clear and accurate?*

Figures were clear and illustrated well. See minor comments on text below:

3. Other minor comments Main Text: p3, line 62; p12, line 132: It doesn't seem necessary or appropriate to cite the dictionary for the definition of robust. p4, line 187: "~20 h" rhythms instead of "~20h-hour" p3, line 70; p5, line 121; p14, line 380; p16, line 416 and p18, line 458: Close parentheses have been doubled in parenthetical references. p14, line 363: "crassa" instead of "Crassa" p17, line 430: "Sigma" instead of "sigma" p18, lines 464 and 483; p20, line 521: put a space between numerical values and units, to be consistent with other entries p19, line 488: "luciferase" instead of Luciferase p20, line 512: "Cell Signaling" instead of "cell signalling" p20, line 526: "single" instead of "Single" Main figures: Fig. 2 p37, line 921: close parenthesis was doubled on "red" Fig. 4 p41, line 989: "0.1 mM" instead of "0.1 mM" for consistency throughout text Supplementary text: line 171: "30

mM HEPES" instead of "30mM HEPES" line 184: "Cell Signaling" instead of "cell signalling" Supplementary figures: Fig. S2A "Drosophila melanogaster" instead of "Drosophila Melanogaster"

3. Significance:

Significance (Required)

This paper revisits the previously proposed idea that rhythmic expression of central TTFL components is not essential for circadian timekeeping to persist. However, this paper does not add a significant advance in the understanding of the underlying reasons behind sustained clock protein rhythmicity like PER in the absence of CRY, since such mechanisms in functional analogs have been shown in other systems, like *Neurospora* (Larrondo et al., 2015). However, this paper does clarify some issues in the field, such as discrepancies between behavioral and cellular rhythms observed in CKO mice, leading future researchers to examine closely the conditions of their CKO rhythmic assays before making conclusions pertaining to rhythmicity. The identification of the kinases as components of the proposed cytosolic oscillator (cytoscillator) needs further validation, but this is perhaps beyond the scope of the paper. The data provides incremental evidence for the existence of a cytoscillator, but opens up opportunities to identify other players, like phosphatases, to establish the connection between the central TTFL and the proposed cytoscillator.

Reviewer #1 (Evidence, reproducibility and clarity (Required)):

****Summary:****

This interesting study by Putker et al. showed that circadian rhythmicity persists in several typical circadian assay systems lacking Cry, including Cry knockout mouse behavior and gene expression in Cry knockout fibroblasts. They further demonstrated weak but significant circadian rhythmicity in Cry- and Per- knockout cells. Cry- (and potentially Per-)-independent oscillations are temperature compensated, and CK1 δ /e still has a role in the period regulation of Cry-independent oscillations.

****Major comments:****

1) The authors propose that the essential role of mammalian Cryptochrome is to bring the robust oscillation. As the authors analyze in many parts, the robustness of oscillation can be validated by the (relative) amplitude and phase/period variation, both of which should be affected significantly by the method for cell synchronization. Unfortunately, the method for synchronization is not adequately written in this version of supplementary information. This reviewer has no objection to the "iterative refinement of the synchronization protocol" but at least the correspondence between which methods were used in which experiments needs to be clearly explained. The detailed method may be found in the thesis of Dr. Wong, but the methods used in this manuscript need to be detailed within this manuscript.

We thank the reviewer for recognising the importance of different synchronisation protocols. In experiments where bioluminescent CKO rhythms were observed, different synchronisation protocols resulted in similar results when comparing WT with CKO cells. The different synchronisation methods used in each experiment are now specified in the supplementary methods.

2) The authors revealed that CKO mice have apparent behavioral rhythmicity under the condition of LL>DD. This is an intriguing finding. However, it should be carefully evaluated whether this rhythmicity (16 hr cycle) is the direct consequence of circadian rhythmicity observed in CKO and CPKO cells (24 hr cycle) because the period length is much different. Is it possible to induce the 16 hr periodicity in CKO mice behavior by 16 hr-L:16 hr-D cycle? Would it be a plausible another possibility that the 16 hr rhythmicity is the mice version of internal desynchronization or another type of methamphetamine-induced-oscillation/food-entrainable-oscillation?

The reviewer makes an excellent suggestion. As described in the manuscript text (page 13), CKO mice have already been shown to entrain to restricted feeding cycles (Iijima et al., 2005) and we therefore assessed whether CKO rhythms would entrain to a 16h day as suggested. Whilst CKO (but not WT) mice showed 16h behavioural rhythms during entrainment, they were arrhythmic under constant darkness thereafter (Revised Figure S2A). CKO cellular rhythms show reduced robustness under constant conditions *ex vivo*, and our other work has revealed that CRY-deficiency renders cells much more susceptible to stress (Wong et al, 2020, BioRxiv). The parsimonious explanation, therefore, is that whilst the cellular timing mechanism remains functional when CRY is absent, the amplitude of cellular clock outputs is severely attenuated (as we showed previously in Hoyle et al., Sci Trans Med,

2017) in a fashion that impairs the fidelity of intercellular synchronisation under most conditions *in vivo*, as well as the molecular mechanisms of entrainment to light-dark cycles.

With respect to the apparent discrepancy between mean periods of CKO cultured cells (~21h), SCN (~19h) and mice (~17h). This is also observed in WT cells (~26h), SCN (~25h) and mice (~24h), simply with a smaller effect size and longer intrinsic period.

We believe this difference in effect size can adequately be explained by differences in oscillator coupling, combined with the reduced robustness of CKO timekeeping. In Figure 1F we show that the range of rhythmic periods expressed by cultured CKO fibroblasts (14-30h) is much greater than for their WT counterparts (range of 22-26h), or that which is observed when cellular oscillators are coupled in CKO SCN (19h). Thus period of CKO oscillations is demonstrably more plastic (less robust) than WT, and with a cell-intrinsic tendency towards shorter period which is revealed more clearly when oscillators are coupled.

In vivo there is more oscillator coupling in the intact SCN than in an isolated slice, from which communication with the caudal and rostral hypothalamus has been removed. Thus it seems plausible that increased coupling *in vivo*, combined with positive feedback via behavioural cycles of feeding and locomotor activity, resonate with a common frequency which is shorter than in isolated tissue.

Critically, for both WT and CKO mice/SCN, the circadian period lies within the range of periods observed in isolated fibroblasts. To communicate this rather nuanced point we have inserted the following text into the supplementary discussion:

“Circadian timekeeping is a cellular phenomenon. Co-ordinated ~24h rhythms in behaviour and physiology are observed in multi-cellular mammals under non-stressed conditions when individual cellular rhythms are synchronised and amplified by appropriate extrinsic and intrinsic timing cues. In light of short period (~16.5h) locomotor rhythms observed in CKO mice after transition from constant light to constant dark, but failure to entrain to 12h:12h light:dark cycles, it seemed plausible that either CKO mice might entrain to an short 8h:8h light:dark (16h day) or else have a general deficiency to entrainment by light:dark cycles. The data in Figure S2 supports the latter possibility, in that neither WT nor CKO mice stably entrained to 16h cycles whereas WT but not CKO mice entrained to 24h days. The bioluminescence oscillations observed in CKO cells conform to the long-established definition of a circadian rhythm (temperature-compensated ~24h period of oscillation with appropriate phase-response to relevant environmental stimuli). Whereas the locomotor rhythms observed in CKO mice under quite specific environmental conditions correlates with both the cellular and SCN data to suggest the persistence of capacity to maintain behavioural rhythms close to the circadian range, but which is masked under most circumstances. We suggest that in vivo the (pathophysiological) stress of CRY-deficiency is epistatic to the expression of daily rhythms in locomotor activity following standard entrainment by light:dark cycles and thus, whilst not arrhythmic, also cannot be described as circadian in the strictest sense.”

3) The authors proposed that CKId/e at least in part is the component of cytosillator (Fig. 5D), and turnover control of PER (likely to be controlled by CKId/e) may be an interaction point between cytosillator and canonical circadian TTFL (Fig. 4). Strictly speaking, this model is not directly supported by the experimental setting of the current manuscript. The contribution of CKId/e is evaluated in the presence of PER by monitoring the canonical

TTFL output (i.e. PER2::LUC); thus it is not clear whether the kinase determines the period of cytosillator. It would be valuable to ask whether the PF and CHIR have the period-lengthening effect on the Nrd1:LUC in the CPKO cell.

Another excellent suggestion, thanks. The experiment, showing similar results in CKO and CPKO cells, was performed and is now reported in Revised Figure S5D. The text was amended as follows: “We found that inhibition of CK1 δ/ϵ and GSK3- α/β had the same effect on circadian period in CKO cells, CPKO cells, and WT controls (Figure 5A, B, S5A, B, D).”

Moreover, our data are further supported by findings in RBCs, where CK1 inhibition affects circadian period in a similar manner as in WT and CKO cells (Beale et al, JBR 2019).

****Minor comments:****

4) The authors argue that the CKO cells' rhythmicity is entrained by the temperature cycle (Fig. 2C). Because the data of CKO cell only shows one peak after the release of constant temperature phase, it is difficult to conclude whether the cell is entrained or just respond to the final temperature shift.

We agree with the reviewer and have replaced the original figure with another recording that includes an extra circadian cycle in free-running conditions (Revised Figure 2C).

5) It would be useful for readers to provide information on the known phenotype of TIMELESS knockout flies; TIM is widely accepted as an essential component of the circadian clock in flies; are there any studies showing the presence of circadian rhythmicity in Tim-knockout flies (even if it is an oscillation seen in limited conditions, such as the neonatal SCN rhythm in mammalian Cry knockout)?

The reviewer is correct that TIM is widely accepted as an essential component of the circadian clock in flies. Using more sensitive modern techniques however, ~50% of classic Tim⁰¹ mutant flies exhibit significant behavioural rhythms in the circadian range under constant darkness, as reported:

<https://opus.bibliothek.uni-wuerzburg.de/frontdoor/index/index/year/2015/docId/11914>

For this reason we employed a full gene knockout of the Timeless gene (Lamaze *et al.*, Sci Rep, 2017), where the majority of flies are behaviourally arrhythmic under constant conditions following standard entrainment by light cycles and therefore represents a more appropriate model for CRY-deficient cells.

We have revised the legend of Figure S2 to include the following:

“N.B. The generation of Tim^{out} flies is reported in Lamaze et al, Sci Rep, 2017. Similar to CRY-deficient mice, whole gene Timeless knockout flies are characterised as being behaviourally arrhythmic under constant darkness following entrainment by light:dark cycles: <https://opus.bibliothek.uni-wuerzburg.de/frontdoor/index/index/year/2015/docId/11914>”

5) Figure 3C shows that the amount of PER2::LUC mRNA changes ~2 fold between time = 0 hr and 24 hr in the CKO cell. This amplitude is similar to that observed in WT cell although the peak phase is different. Does the PER2::LUC mRNA level show the oscillation in CKO cells?

No, we think we have shown convincingly this is not the case. We argue the data in figure 3C show that: (a) there is no circadian variation in mRNA PER2::LUC expression (mRNA levels increase but no trough is observed) and (b) that the temporal relationship between protein and mRNA as observed in WT is broken; i.e. the CRY-independent circadian variation in protein levels cannot be “driven by” changes in transcript levels. Similar results were obtained using transcriptional reporters *Per2*:LUC and *Cry1*:LUC (Figure S3E and F). Moreover, our findings are also in line with previous reports, such as Nangle *et al.* (2014, eLife) and Ode *et al.* (Mol Cell, 2017).

6) Figure 3D: the authors discuss the amplitude and variation (whether the signal is noisier or not) of reporter luciferase expression between different cell lines. However, a huge difference in the luciferase signal can be observed even in the detrended bioluminescence plot. This reviewer concerns that some of the phenotypes of CKO and CPKO MEF reflect the lower transfection efficiency of the reporter gene, not the nature of circadian oscillators of these cell lines.

As reported in the methods, these are stable cell lines rather than transiently transfected cells. The detrended luciferase data presented here do not actually reflect raw levels of luciferase protein expression, but rather reflect the amount of deviation from the 24 hour average. To make it easier to compare expression levels of *Per2*:LUC and *Nr1d1*:LUC between the different cell lines we have added figure S3H, presenting the average raw bioluminescence levels over 24 hours (after 24 hours of recovery from media change; ie from 24-48 hours). Using these data one can appreciate that expression levels of the *Per2* reporter are never lower in CRY KO cells when compared to WT. We hope these data can take away the reviewer’s concerns about expression levels causing the differences observed.

Reviewer #1 (Significance (Required)):

Although Cryptochrome (Cry) has been considered a central component of the mammalian circadian clock, several studies have shown that circadian rhythms are maintained in the absence of Cry, including in the neonate SCN and red blood cells. Thus, although the need for Cry as a circadian oscillator has been debated, its essential role as a circadian oscillator remains established, at least in the cell-autonomous clock driven by the TTFL. This study provides additional evidence that the circadian rhythmicity can persist in the absence of Cry.

More general context, the presence of a non-TTFL circadian oscillator has been one of the major topics in the field of circadian clocks except for the cyanobacteria. In mammals, the authors’ and other groups lead the finding of circadian oscillation in the absence of canonical TTFL by showing the redox cycle in red blood cells (O’Neil, Nature 2011). The presence of circadian oscillation in the absence of *Bmal1* is also reported recently (Ray, Science 2020). *Bmal1* (-CLOCK), CRY, and PER compose the core mechanism of canonical circadian TTFL; thus, this manuscript put another layer of evidence for the non-TTFL circadian oscillation in mammals.

Overall, the manuscript reports several surprising results that will receive considerable attention from the circadian community.

This reviewer has expertise in the field of mammalian circadian clocks, including genomics, biochemistry, and mice's behavior analysis.

Reviewer #2 (Evidence, reproducibility and clarity (Required)):

In the canonical model of the mammalian circadian system, transcription factors, BMAL1/CLOCK, drive transcription of *Cry* and *Per* genes and CRY and PER proteins repress the BMAL1/CLOCK activity to close the feedback loop in a circadian cycle. The dominant opinion was that CRY1 and CRY2 are essential repressors of the mammalian circadian system. However, this was challenged by persistent bioluminescence rhythms observed in SCN slices derived from *Cry*-null mice (Maywood et al., 2011 PNAS) and then by persistent behavior rhythms shown by the *Cry1* and *Cry2* double knockout mice if they are synchronized under constant light prior to free running in the dark (Ono et al., 2013 PLOS One). In the manuscript, the authors first confirmed behavioral and molecular rhythms in the *Cry1/Cry2*- deficient mice and then provided evidence to suggest the rhythms of *Per2*:LUC and *Nr1d1*:LUC in CKOs are generated from the cytoplasmic oscillator instead of the well-studied transcription and translation feedback loop: Constant *Per2* transcription driven by BMAL1/CLOCK plus rhythmic degradation of the PER protein result in a rhythmic PER2 level in the absence of both *Cry1* and *Cry2*, which suggests a connection between the classic transcription- and translation-based negative feedback loops and non-canonical oscillators.

****Major points:****

Line 38-39, "Challenging this interpretation, however, we find evidence for persistent circadian rhythms in mouse behavior and cellular PER2 levels when CRY is absent." The rhythmic behavioral phenotype of *cry1* and *cry2* double knockout mice was first documented by Ono et al., 2013 PLOS ONE, in which eight *cry1* and *cry2* double knockout mice after synchronization in the light displayed circadian periods with different lengths and qualities. The paper reported two period lengths from the *Cry* mutant mice: "An eye-fitted regression line revealed that the mean shorter period was 22.86±0.4 h (n= 8) and the mean longer period was 24.66±0.2 h (n =9). The difference of two periods was statistically significant (p, 0.01).", either of which is quite different from the ~16.5 hr period in Figure 1B of the manuscript. A brief discussion on the period difference between studies will be helpful for readers to understand. Period information from the individual mouse should be calculated and shown since big period variations exist among CKO mice (Ono et al., 2013 PLOS One).

Thanks for this suggestion. The mice used by Ono *et al* were raised from birth in constant light, whereas we used mice that were weaned and raised in normal LD cycles before being subject to constant light then constant dark as adults. Instead of the somewhat subjective fitting of regression lines by eye performed by Ono et al, our analysis was performed using the periodogram analysis routine of ClockLab 6.0 with a significance threshold for rhythmicity of $p=0.0001$. We have now repeated this experiment with 10 adult CKO mice (male and female), and found no evidence for two period lengths in that the second most significant period was consistently double that of the first. As the reviewer suggests, there is a much broader distribution of CKO mouse periods compared with WT, as we also found in cultured cells and SCN. These new data are now reported in revised Figure S1B & C. We have also included a statement about how our study differs from Ono et al in the supplementary discussion.

The behavioral phenotype of *Cry*-null mice and luminescence from their SCNs are robustly rhythmic while fibroblasts derived from these mice only produce rhythms with very low amplitudes compared with those in WT, which may reflect the difference between the SCN's

rhythm and peripheral clocks. The behavioral phenotype is supposed to be controlled mainly by SCN. However, most molecular analyses in the work were done with MEF and lung fibroblasts. These tissues may not be the best representative of the behavioral phenotype of the CKO mice.

Behavioural rhythms of CKO mice are significantly less robust than WT, with mean amplitude less than 50% of WT controls (Figures 1A & B, revised S1B. Furthermore, as reported, 40% of CKO SCN slices exhibited PER2::LUC rhythms, compared with 100% of WT SCN slices (as also observed by Maywood et al., PNAS, 2013), and therefore are also less robust by the definition used in this manuscript.

As now discussed in the revised supplementary discussion:

“Circadian timekeeping is a cellular phenomenon. Co-ordinated ~24h rhythms in behaviour and physiology are observed in multi-cellular mammals under non-stressed conditions when individual cellular rhythms are synchronised and amplified by appropriate extrinsic and intrinsic timing cues.”

The objective of this study was to understand the fundamental determinants that allow mammalian cells to generate a circadian rhythm, which we find does not include an essential role for CRY genes/proteins. Thus the cell is the appropriate level of biological abstraction at which to investigate the phenomenon, whereas the SCN and behavioural recordings simply serve to illustrate the competence of CRY-independent timing mechanisms to co-ordinate biological rhythms at higher levels of biological scale which are manifest under some conditions. To reiterate, the behavioural data supports the cellular observations, not the converse.

Stronger evidence is needed to fully exclude the possibility that in CKO cells, the rhythm is not generated by PERs' compensation for the loss of Crys to repress BMAL1 and CLOCK. Since the rhythms of Per:LUC or Nr1d1:LUC (Figures 3D and S3E) are much weaker than those in WT, molecular analyses might not be sensitive enough to reflect the changes across a circadian cycle in the CKOs if the TTFL still occurs. CLOCK Δ 19 mutant mice have a ~4 hr longer period than WT (Antoch et al., 1997 Cell; King et al., 1997 Cell). CLOCK Δ 19; CKO cells or mice should be very helpful to address the question. Periods of Per:LUC and Nr1d1:LUC from the CLOCK Δ 19; CKO should be similar to those in the CKO alone if the transcription feedback does not contribute to their oscillations.

We agree this would be an interesting experiment, however the data in this manuscript and Wong *et al.* (BioRxiv, 2020), whilst not disputing the existence of the TTFL, strongly suggest that it fulfils a different function to that which is currently accepted and is not the mechanism that ultimately confers circadian periodicity upon mammalian cells. CLOCK Δ 19 is an antimorphic gain-of-function mutation with many pleiotropic effects. Therefore, if the TTFL is not the basis of circadian timekeeping in mammalian cells, it follows that the CLOCK Δ 19 mutation may not elicit its effects on circadian rhythms through delaying the timing of transcriptional activation, as was proposed. As such, whether or not CLOCK Δ 19 alters circadian period of CKO cells/mice would not allow the two models to be distinguished in the way that the reviewer envisions.

Secondly, we cannot detect any interaction between PER2 and BMAL1 in the absence of CRY using an extremely sensitive assay.

Thirdly, very strong biochemical evidence suggests that PER has no repressive function in the absence of CRY (Chiou et al., 2016; Kume et al., 1999; Ode et al., 2017; Sato et al., 2006).

Finally, in several figures particularly 3C and 4A, we show that PER2 peaks at the same time CKO and WT cells, but in CKO cells this is not accompanied by a coincident peak in the mRNA. Thus, even if PER were able to repress BMAL1/CLOCK without CRY, rhythms in PER2 protein level could not be explained by some residual PER/BMAL1-dependent TTFL mechanism.

To address the reviewer's concern however, we have employed mouse red blood cells which offer unambiguous insight into the causal determinants of circadian timing, as we can be absolutely confident that there is no transcriptional contribution to cellular timekeeping. Briefly, we took fibroblasts and RBCs from WT, short period *Tau/Tau* and long period *Afh/Afh* mutant mice. The basis of the circadian phenotype of these mutations is quite well established as occurring through the post-translational regulation of PER and CRY proteins respectively, and result in short and long period PER2::LUC rhythms compared with WT fibroblasts. RBCs do not express PER or CRY proteins, and commensurately no genotype-dependent differences of RBC circadian period were observed (Beale et al, 2020, in submission). In contrast, RBC circadian rhythms are sensitive to pharmacological inhibition of casein kinase 1 (Beale et al., JBR, 2019).

Lines 51-52, "PER/CRY-mediated negative feedback is dispensable for mammalian circadian timekeeping" and lines 310-311, "We found that transcriptional feedback in the canonical TTFL clock model is dispensable for cell-autonomous circadian timekeeping in animal and cellular models." The authors have not excluded the possibility that the rhythmic behaviors of the CKO mice are derived from the PERs' compensation for the role of Crys in the feedback loop of the circadian clock in the SCN. In the fibroblasts, only two genes, *Per2* and *Nr1d1*, have been studied in the work, which cannot be simply expanded to the thousands of circadian controlled genes. Also amplitudes of PER2:LUC and NR1D1:LUC in the CKOs are much lower than those in WT and no evidence has been provided to show that their weak rhythms are biologically relevant.

The definition of a circadian rhythm (Pittendrigh, 1960) does not mention biological relevance or stipulate any lower threshold for amplitude. As now stated in the revised text (page 6):

“PER2::LUC rhythms in CKO cells were temperature compensated (Figure 2A, B) and entrained to 12h:12h 32°C:37°C temperature cycles in the same phase as WT controls (Figures 2C), and thus conform to the classic definition of a circadian rhythm (Pittendrigh, 1960) – which does not stipulate any lower threshold for amplitude or robustness.”

We make no claims about biological relevance or amplitude in this manuscript, which are addressed in our related manuscript (Wong *et al.*, BioRxiv, 2020). In this related manuscript, we explicitly address whether CRY is necessary for mammalian cells to maintain a circadian rhythm in the abundance of clock-controlled proteins and find that it is not. Indeed, twice as many rhythmically abundant proteins are observed in CKO cells than WT controls, which suggests that, if anything, CRY functions to suppress rhythms in protein abundance rather than to generate them.

We observe circadian rhythms in the activity of two different bioluminescent reporters, which have already been extensively characterised. The mouse and SCN data in figure 1 are correlative, and simply show that previous published observations are reproducible. PER2::LUC oscillations are not accompanied by *Per2* mRNA oscillations. This, together with the absence of a BMAL1-PER2::LUC complex strongly argues against a model where PER2 oscillations are driven by residual (PER2-driven) transcriptional oscillations.

We therefore concede the reviewer's point that we "cannot exclude rhythmic behaviors of the CKO mice are derived from the PERs' compensation for the role of Crys in the feedback loop of the circadian clock in the SCN". The reviewer will agree however, that there exists very strong biochemical evidence suggests that PER has no repressive function in the absence of CRY (Chiou et al., 2016; Kume et al., 1999; Ode et al., 2017; Sato et al., 2006); that there exists no experimental evidence to suggest that PERs can fulfil this function in the absence of CRY in any mammalian cellular context; and finally that our observations are not consistent with the canonical model for the generation of circadian rhythms in mammals.

We have therefore amended the text to focus on CRY specifically, as follows:

"~~PER~~/CRY-mediated negative feedback is dispensable for mammalian circadian timekeeping"

Page 12. *"We found that CRY-mediated transcriptional feedback in the canonical TTFL clock model is dispensable for cell-autonomous circadian timekeeping in cellular models. Whilst we cannot exclude the possibility that in the SCN, but not fibroblasts, PER alone may be competent to effect transcriptional feedback repression in the absence of CRY, we are not aware of any evidence that would render this possibility biochemically feasible."*

****Minor points:****

Lines 66-67, "... (Dunlap, 1999; Reppert and Weaver, 2002; Takahashi, 2016)." to "... (reviewed in Dunlap, 1999; Reppert and Weaver, 2002; Takahashi, 2016)."

Thanks, changed as requested.

Line 70, "... (Liu et al., 2008..." to "... (Liu et al., 2008..."

Thanks, changed as requested.

Lines 174-175, "Considering recent reports that transcriptional feedback repression is not absolutely required for circadian rhythms in the activity of FRQ...". Larrondo et al., 2015 paper says "however, in such Δ fwf-1 cells, the amount of FRQ still oscillated, the result of cyclic transcription of *frq* and reinitiation of FRQ synthesis." The point of the paper is "we unveiled an unexpected uncoupling between negative element half-life and circadian period determination." instead of "...transcriptional feedback repression is not absolutely required for circadian rhythms in the activity of FRQ,"

This is a good point which, following discussion with Profs Dunlap and Larrondo, we have revised into *"no obligate relationship between clock protein turnover and circadian regulation of its activity"* – a more accurate summary of their findings.

Lines 249-252, "CKO cells exhibit no rhythm in Per2 mRNA (Figure 3C, D), nor do they show a rhythm in global translational rate (Figure S4A, B), nor did we observe any interaction between BMAL1 and S6K/eIF4 as occurs in WT cells (Lipton et al, 2015) (Figure S4C)." In figures 3D and S3E, in CKO and CPKO cells the Per2:LUC data without fitting look better than that of Nr1d1:LUC. But the Nr1d1:LUC rhythm became clear after fitting the raw data. So to better visualize the low amplitude rhythm, if any, of Per2:LUC and compare with Nr1d1:LUC, fitted the Per2:LUC data in CKOs and CPKOs in Figure 3D and S3E should be shown as what has been done to Nr1d1:LUC.

Thanks, these data can be found in Figure S3F. The detrended Per2:Luc CKO and CPKO bioluminescence traces were better fit by the null hypothesis (straight line) than a damped sine wave ($p > 0.05$) and so were not significantly rhythmic by the criteria used in this manuscript.

Lines 258-259, "much less than the half-life of luciferase expressed in fibroblasts under a constitutive promoter" In figure S4D, the y-axis of the PER2::LUC is ~800 while the y-axis of the SV40::LUC is ~600000. The over-expressed LUC by the SV40 promoter might saturate the degradation system in the cell so the comparison is not fair. A weaker promoter with the level similar to Per2 should be used to make the comparison.

Thank you for this suggestion. In our experience, the SV40 promoter is actually a rather weak promoter compared with CMV, and faithfully facilitates the constitutive (non-rhythmic) expression of heterologous proteins such as Luciferase (Feeney et al., JBR, 2016). It has been shown previously that constitutive over-expression of heterologous proteins such as GFP or even CRY1 does not affect circadian rhythms in fibroblast cells (e.g. Chen et al., Mol Cell, 2009). To address the reviewer's reasonable concern however, multiple stable SV40:Luc fibroblast lines were generated by puromycin selection, grown to confluence in 96-well plates, then treated with 25 $\mu\text{g}/\text{mL}$ CHX at the beginning of the recording. Random genomic integration of SV40:Luc leads to a broad range of different levels of luciferase expression, evident from the broad range of initial luciferase activities. For each line the decline in luciferase activity was fit with a simple one-phase exponential decay curve ($R^2 \geq 0.98$) to derive the half-life of luciferase in each cell line. There was no significant relationship between the level of luciferase expression and luciferase stability (straight line vs. horizontal line fit p -value = 0.82). Therefore constitutive expression of SV40:Luc in fibroblasts does affect the cellular protein degradation machinery within the range of expression used for our half-life measurements. These new data are reported in Revised Figure S3H.

Line 430, "sigma" to "Sigma".

Changed

In figure S2, the classification of rhythms in Drosophila is not clear since even the "Robustly rhythmic" ones have high background noise. Detrending or fitting the data might be able to improve the quality of the rhythms prior to classification.

These are noisy data as they come from freely behaving flies. The mean data was shown in Figure S3A and individual examples in S3B, and look very similar to previous bioluminescence fly recordings of XLG-LUC flies in papers from the Stanewsky lab who have published extensively using this model. The classifications arose from double-blinded

analysis of the bioluminescence traces by several individuals, but we agree that this was not clearly communicated in our original submission. In Revised figure S2 we now present the mean bioluminescence traces, with and without damped sine wave vs. straight line fitting, as suggested, which is more consistent with the mammalian cellular data presented elsewhere.

In figure S3B, the original blots for Per2 including Input and IP should be shown.

The original blots for BMAL1 are shown in figure S3I. PER2::LUC levels were assessed by measuring bioluminescence levels present on the anti-bmal1-beads, as described in the figure 3B legend.

Supplemental information

Line 44, "...(reviewed in (Lakin-Thomas,...)" to "...(reviewed in Lakin-Thomas,..."

Changed

Line 188, "Period CDS", the full name of CDS should be provided the first time it appearances.

Changed to "coding sequence".

Reviewer #2 (Significance (Required)):

The work suggests a link between the TTFL and non-canonical oscillators, which should be interesting to the circadian field.

Reviewer #3 (Evidence, reproducibility and clarity (Required)):

Summary:

The paper "CRYPTOCHROMES confer robustness, not rhythmicity, to circadian timekeeping" by Putker et al. answers the question of whether or not the rhythmic abundance of clock proteins is a prerequisite for circadian timekeeping. They addressed this by monitoring PER2::LUC rhythms in WT and CRY KO (CKO) cells. CRY forms a complex with PER, which in turn represses the ability of CLOCK/BMAL1 to drive the expression of clock-controlled genes, including PER and CRY. Consistent with previous observations, the authors found residual PER2::LUC rhythms in CKO SCN slices, fibroblasts and in a functional analogue KO of CRY in *Drosophila*, even in the absence of rhythmic Per2 transcription due to the loss of CRY as a negative regulator of the oscillation. They have shown that these rhythms, in the absence of CRY, follow the formal definition of circadian rhythms. They attributed these residual PER2::LUC rhythms to the maintenance of oscillation in PER2::LUC stability independent of CRY, by testing the decay kinetics of luciferase activity when translation is inhibited. Moreover, they implicated the kinases CK1 δ/ϵ and GSK3 to be involved in regulating PER2::LUC post-translational rhythms through kinase inhibitor studies. They concluded that CRY is not necessary for maintaining PER2::LUC rhythms, but plays an important role in reinforcing high-amplitude rhythms when coupled to a proposed "cytosillator" likely composed of CK1 δ/ϵ and GSK3.

****Major comments:****

The authors have shown sufficient data that under different testing conditions (mice locomotor activity, SCN preps or fibroblasts), behavioral rhythms and PER2::LUC rhythms are still observed in the CRY KO (CKO) cells, contrary to a previous study (Liu et al., 2007). They also indicated limitations to some of the experimental work. However, there are some parts of the paper that need clarification to support their conclusions.

1. In Fig. 1A, the x-axes of the actograms for WT and CKO are different. While they mentioned this in the figure legend, and described the axis transformation in Fig. S1A, they need a justification statement about why they did this in the results.

Thanks, we have included the following sentence in the results section as requested: *“Figure 1 representative actograms are plotted as a function of endogenous tau (τ) to allow the periodic organisation of rest-activity cycles to be readily discerned; 24h-plotted actograms are shown in Figure S1A and S2A”*

2. In an attempt to show conservation of their proposed role for CRY, they tested the model system *Drosophila melanogaster* where TIMELESS serves as the functional analogue of CRY. While they showed in the figures and described in the text that rhythms still persisted with lower relative amplitude in the TIMELESS-deficient flies, they did not describe any period differences between WT and mutant. Showing the period quantification in Supp. Fig. S2 using the robustly rhythmic datasets, and describing this data in the text, will strengthen their claim.

These analyses are now reported in revised Figure S2 as requested. As described in our response to reviewer 2, the “robustly rhythmic” flies were scored as such through double-blinded analysis by several individuals. We hope the reviewer will appreciate our concern that exclusion of the majority of TIMELESS-deficient flies that were not robustly rhythmic might skew their apparent period by unconscious bias towards favouring traces that most clearly resemble robustly rhythmic WT controls. To avoid any potential bias we therefore included all flies of both genotypes in the analysis of circadian period for the revised figure, as suggested by our other reviewers.

In Fig. S2B, there is no clear distinction between the representative datasets shown for poorly rhythmic and arrhythmic, i.e. they all appear arrhythmic, without an indicated statistical test. The authors could present better representative data to better reflect the categories.

As described above, we now show the grouped mean with and without fitting for all flies of both genotypes. The statistical test for rhythmicity and analysis of circadian period is now the same as was performed for the cellular data presented elsewhere.

3. In Fig. 2A, the authors note the lack of rhythmicity in the CKO fibroblasts in the 1st three days at 37°C. How are the conditions here different from fibroblasts in Fig. 1E, where rhythms are seen during the 1st three days in CKO fibroblasts?

As discussed in the manuscript, PER2::LUC rhythms in CKO cells and SCN are observed stochastically between recordings i.e. if one dish in a recording showed rhythms, all dishes showed rhythms and vice versa. The media change that occurred after 3 days in Fig 2A, in

this case, was sufficient to initiate clear rhythms of PER2::LUC in all experimental replicates. In other experiments, media change did not have this effect. Herculean efforts by multiple lab members over many years, including the PI, have been unable to delineate the basis of this variability – which is discussed at length in the thesis of Dr. David Wong <https://www.repository.cam.ac.uk/handle/1810/300610>. As such, we clearly state in the discussion:

“We were unable to identify all of the variables that contribute to the apparent stochasticity of CKO PER2::LUC oscillations, and so cannot distinguish whether this variability arises from reduced fidelity of PER2::LUC as a circadian reporter or impaired timing function in CKO cells. In consequence, we restricted our study to those recordings in which clear bioluminescence rhythms were observed, enabling the interrogation of TTFL-independent cellular timekeeping.”

4. The authors claimed in the results section- "in contrast and as expected, Per2 mRNA in WT cells varied in phase with co-recorded PER2::LUC oscillations." but Fig. 3C does not show this expected lag between mRNA and protein levels. This needs to be explained

No lag is expected *in vitro*. A lag between PER protein levels and Per mRNA does occur *in vivo* and is very likely to be attributable to daily rhythms in feeding (Crosby et al, Cell, 2019), where increased insulin signalling elicits an increase in PER protein production 4-6h after E-box and GRE-stimulated increase in Per transcription.

When luciferin is saturating intracellularly, PER2::LUC activity correlates most closely with the amount of PER2::LUC protein that was translated during the preceding 1-2h, rather than the total amount of PER2, due to the enzymatic inactivation of the luciferase protein (Feeney et al, JBR, 2016). Consistent with many previous observations, under constant conditions, the rate of nascent PER protein synthesis is largely determined by the level of Per2 mRNA, and thus more similar phases are observed between protein and mRNA *in vitro* than *in vivo*.

We have inserted an additional citation of Feeney et al at this point in the text to make this clear.

5. In Figs. 5A-B, the PER2::LUC periods in the CKO untreated cells seem to vary significantly between A, B, and C. While this could be due to the high variability in the rhythms that were previously described by the authors, the average periods here seem to be longer than the one reported in Fig. 1F. Are there specific condition differences?

There are no specific condition differences. As reported in Figure S1B, D & E, the range of CKO cellular periods is simply much broader than for WT cells. Over several dozen experiments the average period was significantly shorter, but the period variance is an equally striking feature of rhythms in these cells which we take as evidence for their lack of robustness.

Would additional experiments be essential to support the claims of the paper?

1. There is sufficient experimental data to support the major claims; however some suggested experiments are listed below.

a. If CKO exhibits residual rhythms in PER::LUC, it would be interesting to know how CRY overexpression influences PER2::LUC rhythms, or point to previous reference papers which

may have already shown such effects. The prediction would be PER2::LUC levels will still be rhythmic when CRY is overexpressed. What would be the extent of "robustness" conferred by CRY on PER2::LUC rhythms based on CRY KO and overexpression studies?

These experiments have largely already been performed (see Chen et al., Mol Cell; Nangle et al., eLife, 2014; Fan et al., Curr Biol, 2007; Edwards et al., PNAS, 2016) and are cited in this manuscript. As suggested, PER2 rhythms remain intact under CRY1 over-expression, though are clearly perturbed, but their robustness was not investigated in any detail. We hope to be able to address this important question in our subsequent work.

b. The authors found that CK1 δ/ϵ and GSK3 contribute to CRY-independent PER2 oscillations by showing that addition of kinase inhibitors affect the PER2::LUC period lengths in WT and CKO in the same manner. It would be interesting to know if a) PER2::LUC stability and b) PER2 phosphorylation status, is affected in WT and CKO in the presence of the inhibitors, or point to previous reference papers which may already have shown such effects.

As the reviewer points out, PER2 stability is already reported to be regulated via phosphorylation by GSK3 and CK1. We have made explicit reference to this in the revised manuscript as follows:

“In contemporary models of the mammalian cellular clockwork CRY proteins are essential for rhythmic PER protein production, however, the stability and activity of PER proteins are also regulated post-translationally (Lee et al., 2009; Philpott et al., 2020; Iitaka et al., 2005).”

Are the data and the methods presented in such a way that they can be reproduced?

1. The protocol for the inhibitor treatments are not in the main or supplemental methods.

In the main text methods, section luciferase recordings we state: *“For pharmacological perturbation experiments (unless stated otherwise in the text) cells were changed into drug-containing air medium from the start of the recording. Mock-treatments were carried out with DMSO or ethanol as appropriate.”*

Are the experiments adequately replicated and statistical analysis adequate?

1. All experiments had the sufficient number of technical and biological replicates to make valid statistical analyses. For Fig. S2, the authors used RAIN to assess rhythmicity in WT and mutant flies, but it is not clear whether the different categories (rhythmic, poorly rhythmic, and arrhythmic) were based on amplitude differences alone, or a combination of amplitude and p-values as determined by RAIN.

As reported above, we have revised the analysis of the fly data to be consistent with the cellular data reported elsewhere in the manuscript.

Minor comments:

1. Are prior studies referenced appropriately?

Authors may wish to include Fan et al., 2007, Current Biology which demonstrated that

cycling of CRY1, CRY2, and BMAL1 is not necessary for circadian-clock function in fibroblasts.

Apologies for the omission of citation to this excellent paper. Now referenced in the introduction.

2. Are the text and figures clear and accurate?

Figures were clear and illustrated well. See minor comments on text below:

3. Other minor comments

Main Text:

p3, line 62; p12, line 132: It doesn't seem necessary or appropriate to cite the dictionary for the definition of robust.

Thanks for this suggestion. During preparation of the manuscript we found that there was some disagreement between authors as to the meaning of robustness in a circadian context. We therefore feel it most necessary to define clearly what we mean by the use of this word to avoid any potential ambiguity.

p4, line 187: "~20 h" rhythms instead of "~20h-hour"

p3, line 70; p5, line 121; p14, line 380; p16, line 416 and p18, line 458: Close parentheses have been doubled in parenthetical references.

p14, line 363: "crassa" instead of "Crassa"

p17, line 430: "Sigma" instead of "sigma"

p18, lines 464 and 483; p20, line 521: put a space between numerical values and units, to be consistent with other entries

p19, line 488: "luciferase" instead of Luciferase

p20, line 512: "Cell Signaling" instead of "cell signalling"

p20, line 526: "single" instead of "Single"

We thank the reviewer for his/her thoroughness, all of the above have been changed.

Main figures:

Fig. 2 p37, line 921: close parenthesis was doubled on "red"
This was actually correct.

Fig. 4 p41, line 989: "0.1 mM" instead of "0.1 mM" for consistency throughout text

Supplementary text:

line 171: "30 mM HEPES" instead of "30mM HEPES"

line 184: "Cell Signaling" instead of "cell signalling"

Supplementary figures:

Fig. S2A "Drosophila melanogaster" instead of "Drosophila Melanogaster"

All of the above have been changed.

Reviewer #3 (Significance (Required)):

This paper revisits the previously proposed idea that rhythmic expression of central TTFL components is not essential for circadian timekeeping to persist. However, this paper does not add a significant advance in the understanding of the underlying reasons behind sustained clock protein rhythmicity like PER in the absence of CRY, since such mechanisms in functional analogs have been shown in other systems, like Neurospora (Larrondo et al., 2015). However, this paper does clarify some issues in the field, such as discrepancies between behavioral and cellular rhythms observed in CKO mice, leading future researchers to examine closely the conditions of their CKO rhythmic assays before making conclusions pertaining to rhythmicity. The identification of the kinases as components of the proposed cytosolic oscillator (cytosillator) needs further validation, but this is perhaps beyond the scope of the paper. The data provides incremental evidence for the existence of a cytosillator, but opens up opportunities to identify other players, like phosphatases, to establish the connection between the central TTFL and the proposed cytosillator.

Thank you for submitting your revised Review Commons manuscript for consideration by The EMBO Journal. In light of the reviewer comments and the interest of the subject of the study, I have sent it back to reviewer #1 for the assessment of your responses to their originally raised concerns. As you will see from the comments copied below, the reviewer is in principle satisfied with the scope of the revision, but also requests softening the interpretation of the observed behavioural rhythms in Cry DKO mice. I would therefore be happy to accept the manuscript for publication in The EMBO Journal after incorporation of these final comments and reformatting of the manuscript according to The EMBO Journal guidelines (<https://www.embopress.org/page/journal/14602075/authorguide> and the points below). According to the journal's style, please also include the "Supplemental discussion" section in the main manuscript file.

Please feel free to contact me if you have any further questions regarding the revision. Thank you for the opportunity to consider your work for publication. I look forward to receiving your revised manuscript.

Referee #1:

The authors have adequately addressed all concerns raised by this reviewer. The revised Figure 5D supports the presence of CKI/GSK3-sensitive oscillator independent from the CRY-PER complex. For the mice 16 hr behavioral rhythms, unfortunately, this reviewer feels that the revised data still does not fully support the connection between CKO/CPKO cellular rhythms and mice behavioral rhythms. The supplemental discussion concerns this point. However, the summary sentence "we find evidence for persistent circadian rhythms in mouse behavior" implies that the Cry DKO mice shows circadian rhythmicity (in a strict sense). This reviewer suggests changing this sentence to more accurately reflect the fact that "the locomotor rhythms were observed in CKO mice under quite specific environmental conditions."

Thank you for submitting a revised version of your manuscript. I am afraid that several editorial and formatting issues still have to be addressed before we can forward your manuscript to our publishers.

- 1) I noticed that some figure panels are reused in the manuscript: Fig. 1A in EV1A, Fig. EV3E in Fig. EV3F. Please indicate this in the figure legends.
- 2) Please rename "Experimental Procedures" into "Materials and Methods".
- 3) Please assemble Figure EV3 in a single A4 portrait format image or split into two images - we can extend the number of EV figures to six if needed.
- 4) We require a Data Availability Section at the end of Materials and Methods (you can find more information here: <https://www.embopress.org/page/journal/14602075/authorguide#dataavailability>). As far as I can see, no data deposition in external databases is needed for this paper. If I am correct, then please state in this section: This study includes no data deposited in external repositories.
- 5) The funding information differs between our online submission system and the manuscript file (e.g., National Institutes of Health (GM118102) is missing in our online system). Please compare and extend as necessary.
- 6) Please add a short Table of Contents in front of the Appendix.
- 7) We generally encourage publication of source data, in particular for electrophoretic gels and blots, with the aim of making primary data more accessible and transparent to the reader. We would need one file per figure (which can be a composite of source data from several panels), uploaded as "Source data files". The gels should be labeled with the appropriate figure/panel number, and should have molecular weight markers; further annotation would clearly be useful but is not essential. These files will be published online with the article as supplementary "Source Data". Please let me know if you have any questions about this policy.
- 8) Papers published in The EMBO Journal are accompanied online by a 'Synopsis' to enhance discoverability of the manuscript. It consists of A) a short (1-2 sentences) summary of the findings and their significance, B) 2-3 bullet points highlighting key results and C) a synopsis image that is 550x300-600 pixels large (width x height, jpeg or png format). You can either show a model or key data in the synopsis image. Please note that the size is rather small and that text needs to be readable at the final size. Please send us this information along with the revised manuscript.

Our publisher is currently performing their pre-publication check of data information provided in figure legends, and I will send you their comments once the check has been completed to allow you to already go ahead with the preparation of the revision.

I have checked with our publishers regarding the requested NLM manuscript ID. Our colleagues at Wiley have informed me that we can provide a Pubmed ID once the article is published online.

Please let me know if you have any further questions regarding any of these points. You can use the link below to upload the revised files.

The authors performed the requested changes.

Editor accepted the revised manuscript

Corresponding Author Name: John S. O'Neill

Journal Submitted to: The EMBO Journal

Manuscript Number: EMBOJ-2020-106745